# LARGE LANGUAGE MODELS AS WINDOWS ON THE MENTAL STRUCTURE OF PSYCHOPATHOLOGY

## ABSTRACT

How people represent the world determines how they act on it, as these internal representations bias what information is retrieved from memory, the inferences that are made and which actions are preferred. The structure of these representations are built through experience by extracting relevant information from the environment. Recent research has demonstrated that representational structure can also respond to the internal motives of agents, such as their aversion to uncertainty, which impacts their behavior. This opens the possibility to directly target internal structures to cause behavioral change in psychopathologies, one of the tenets of cognitive-behavioral therapy. For this purpose, it is crucial to understand how internal structures differ across psychopatologies. In this work, we show that Large Language Models (LLMs) could be viable tool to infer structural differences linked to distinct psychopathologies. We first demonstrate that we can reliably prompt LLMs to generate (verbal) behavior that can be detected as psychopathological by standard clinical assessment questionnaires. Next, we show that such prompting can capture correlational structure between the scores of diagnostic questionnaires observed in human data. We then analyze the lexical output patterns of LLMs (a proxy of their internal representations) induced with distinct psychopathologies. This analysis allows us to generate several empirical hypotheses on the link between mental representation and psychopathologies. Finally, we illustrate the usefulness of our approach in a case study involving data from Schizophrenic patients. Specifically, we show that these patients and LLMs prompted to exhibit behavior related to schizophrenia generate qualitatively similar semantic structures. We suggest that our novel computational framework could expand our understanding of psychopathologies by creating novel research hypotheses, which might eventually lead to novel diagnostic tools. [1]

## 1 INTRODUCTION

Uncovering internal structure is crucial for properly understanding how the mind works (Johnson-Laird, 1980; Brady et al., 2011; Osgood et al., 1957). Through the joint analysis of neural network model simulations and empirical behavioral studies, seminal works in the early 80's (McClelland et al., 1987; Rumelhart et al., 1986; Hopfield, 1982) have revealed key insights regarding representation learning and structure of mental processes. Since then, this tradition has carried on and modern models nowadays provide a rich source of information to uncover the structure of mental representation subtending complex human (Ito et al., 2022; Caucheteux et al., 2023) and animal (Recanatesi et al., 2022; Sohn et al., 2019) behavior, while also providing insights for the development of AI algorithms (Hassabis et al., 2017).

Research on the impact of psychopathologies on thinking and reasoning has focused on cognitive processes. Several deficits or biases have been observed in memory, lexical processing, perception, or decision-making (Halligan & David, 2001). In contrast, the structure of the contents of the mind have been for the most part neglected. Internal structure defines how information (e.g., lexical representations) is encoded mentally and in high-dimensional neural activity space, with crucial consequences for behavior. Even though changing structure is one of the basis of cognitive-behavioral therapy, its link with psychopathology has been neglected (Arntz, 2020). However, directly accessing

---

[1]Code available at https://anonymous.4open.science/r/LLM-and-Psychopathology-5323

internal structure via neural recordings has a significant temporal and financial cost. An indirect method is analyzing the patterns of lexical outputs of the mental structure in healthy individuals (Vives et al., 2023) and psychopathology (Nour et al., 2023). This opens the avenue for Large Language Models (LLMs) to be used for uncovering the internal structures linked to psychopathology, since LLMs not only excel precisely at language use, but their embeddings correlate highly with semantic similarity judgments Marjieh et al. (2024); Gatto et al. (2023).

Interestingly, it has recently been argued that Large Language Models (LLMs) can be used as tools to understand human cognition (Frank, 2023). In fact, recent studies have focused on evaluating whether LLMs have the ability to generate human cognitive abilities by evaluating their (natural language) behavioral patterns (Dasgupta et al., 2022; Herrera-Berg et al., 2023; Shiffrin & Mitchell, 2023; Binz & Schulz, 2023; Kosinski, 2023; Mahowald et al., 2023; Le Mens et al., 2023; Hu et al., 2023; Ullman, 2023). However, ***whether LLMs can be leveraged to uncover new hypotheses on how mental structure is affected by psychopathology remains an open question.*** If verified, such an application of LLMs could significantly speed the scientific inquiry of how structure is linked to psychopathologies. Indeed, analyzing the lexical output of LLMs prompted to behave with a given psychopathology, and finding how representations vary across psychopathologies, could optimally guide researchers towards exploring (and potentially confirming) the mental structures of human psychopathology.

In this work, we propose a computational framework that allows the use of LLMs as a test-bed of psychopathology and thereby probe and compare how the structure of mental representations could be affected by psychopathology (Figure 1). Our framework allows us to select an LLM, a specific psychopathology, a prompt method, and probe the structure of mental representations. The contribution of this work is fourfold:

1. We demonstrate that we can reliably prompt LLMs to generate (verbal) behavior that can be detected as psychopathological by standard clinical assessment questionnaires[2].

2. We uncover the correlational structure between distinct psychopathology-induced LLMs with respect to scores that evaluate psychopathologies based on well-defined and validated questionnaires; and compare each induced LLM with human data.

3. We demonstrate that inducing different psychopathologies in LLMs leads to distinct structures in semantic representations.

4. We shed light on the potential use of LLMs to study the mental structure associated with psychopathology.

## 2 RELATED WORK

**Probing the mental structure of psychopathology.** Whereas several studies evaluate the how psychopathology affects cognitive processes such as memory, attention, and decision-making (Wiers et al., 2013), only a few studies evaluate their associated mental representations, despite their crucial implications for treatments (Arntz, 2020) and diagnosis (Hyman, 2010). One such study investigated semantic representations in schizophrenic patients (Lundin et al., 2020) who passed a verbal fluency task. In this task, participants generate as many words as possible with respect to a given category (i.e., animals). This study found that compared with healthy controls, schizophrenics generate words that are farther to one another in semantic space, as measured by the cosine similarity between word2vec embeddings (also see (Nour et al., 2023) for a similar effect). In the same vein, but focusing on personality features rather than psychopathology, recent work has shown that humans displaying high levels of uncertainty aversion (a trait closely linked to anxiety disorders (McEvoy & Mahoney, 2012)) represent words in an expanded semantic space (Vives et al., 2023). Uncertainty-averse people would thereby reduce semantic interference at the expense of generalization abilities. In fact, probing the structure of mental representations via the analysis free associations is a widely used approach (Aeschbach et al., 2024; De Deyne et al., 2019), and will be resorted to here as well.

**LLMs as models of pathological mental representation structure.** Given the recent impressive skills displayed by LLMs (Bubeck et al., 2023), cognitive science researchers have sought out the

---

[2]For simplicity, in the remainder of the text we refer to this prompting as *inducing* a given psychopathology.

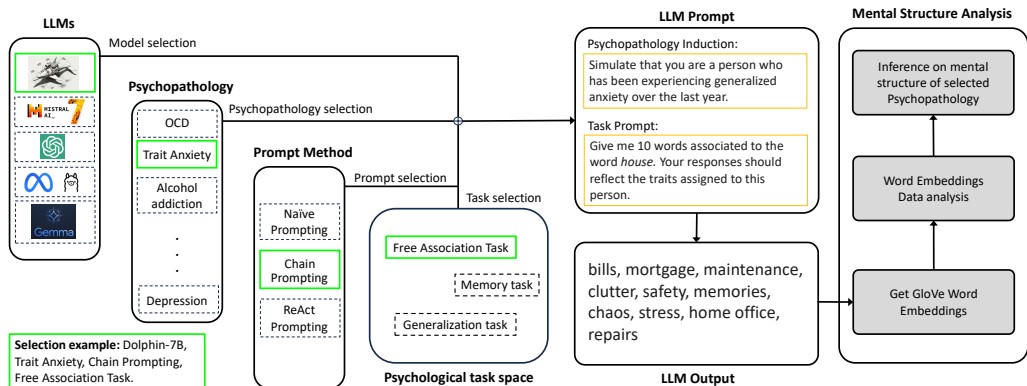

Figure 1: Computational framework. One selects an LLM (e.g., dolphin-7B), a psychopathology of interest (e.g., Trait Anxiety), a prompt method (e.g., PT-prompting), and a psychological task (e.g., free association task). The prompt first induces the psychopathology of interest, and then describes the psychological task to perform under this induction for each LLM. The output words of the LLMs are then passed through an LLM-agnostic word embedding (GloVe) to be analyzed and infer properties of the mental structure associated with the prompted psychopathology.

possibility of using LLMs as cognitive models (Frank, 2023). However, to our knowledge, two studies relate LLMs with psychopathology (Kambeitz et al., 2023; Coda-Forno et al., 2023). One of these studies tackles a different question than the one evaluated in this work, and focuses on whether the psychological concepts present in questionnaires that evaluate psychopathology (see below) are represented similarly in psychopathological patients and LLMs (Kambeitz et al., 2023). The other study induces anxiety in GPT-3.5 and evaluates the decision-making profile of such induction. The authors observed that in addition to scoring highly on questionnaires that evaluate anxiety, mood (anxious or happy) induction modulates exploratory behavior in the decision-making task. Related to our proposal, a recent study suggests that distinct personality types can be induced in LLMs (Jiang et al., 2024). Importantly, we extend their approach to the reliable induction of psychopathology, crucially allowing us to use LLMs as windows on the mental structure of psychopathology. This extension is of great significance as it allows us to predict unexplored mental structures of many psychopathologies. Therefore, aside from gaining fundamental knowledge of pathological mental structure, our work lays a testbed for future empirical studies in this field.

**Relation to Computational Psychiatry.** An emergent field lying in the intersection between computational cognitive (neuro)science and clinical psychiatry is that of computational psychiatry (Huys et al., 2016). Broadly speaking, this field uses data-driven and computational modeling approaches to, respectively, improve diagnostic (Silva et al., 2014) or treatments (Gordon et al., 2015) and investigate the underlying cognitive processes giving rise to psychopathological behavior (Browning et al., 2015; Gold et al., 2012) or neural patterns (Maia & Frank, 2011; Murray et al., 2014; Maia & Cano-Colino, 2015). Our work could add an important branch to the field of computational psychiatry by generating a framework to predict a variety of mental structures associated with certain psychopathologies, and thereby define a psychological task space that can improve the efficiency of data-driven diagnostic machine-learning models.

**Persona prompting.** More generally, our work relates to a growing body of research investigating the effects of LLM impersonification to reveal socio-cultural biases (Gupta et al., 2023), replicate economic, psycholinguistic, and social psychology experiments (Aher et al., 2023), or simulate large scale social computing prototypes (Park et al., 2022).

## 3 INDUCING AND EVALUATING PSYCHOPATHOLOGY IN LLMS

We evaluated five LLMs: dolphin-7B, mistral-7B (Jiang et al., 2023), gpt-3.5-turbo, llama-2 (Touvron et al., 2023), and gemma (based on Team et al. (2023)). Each of these models was evaluated with

three temperature values: 0.3, 0.7, and 0.9. For the sake of brevity, in the remainder of the article, we report LLM results with a temperature value of 0.3 (except if mentioned otherwise), as this value generated results that are closer to human behavioral patterns (see section 3.2). We refer the reader to the appendix for results with other temperature values.

## 3.1 PSYCHOPATHOLOGY INDUCTION

We tested three prompting methods to separately induce[3] nine types of psychopathology: Depression, trait anxiety (TA), eating disorder (ED), alcohol addiction (AA), impulsivity, schizophrenia, obsessive-compulsive disorder (OCD), apathy, and social anxiety (SA). We chose these disorders as they were evaluated in Gillan et al. (2016), thereby allowing us to compare the LLMs evaluation to those of humans. We described our prompting techniques in what follows (prompting examples for all methods are available in table 1, appendix A.1).

**Naive prompting.** The Naive prompting method was similar to that implemented in Jiang et al. (2024). We prompt models as follows: "Simulate that you are a person who has been experiencing $X$ over the last year"; with $X \in \{depression, TA, ED, AA, impulsivity, schizophrenia, OCD, apathy, SA\}$.

**Chain prompting.** Our Chain prompting method is inspired by the original work of Jiang et al. (2024), and combines it with the idea of psychological traits (PTs). In a single prompt, we jointly generate the Naive prompt, a psychological traits profile prompt: "You are a $\{trait_1\}, ..., \{trait_N\}$", and a psychopathology vignette (PV) prompt: "This person $\{pathology\_vignette\}$". The PTs were directly extracted from the DSM-5 (American Psychiatric Association et al., 2013), and PVs were constructed using gpt-3.5-turbo by prompting the model with the PTs and examples of vignettes, in order to generate psychopathology-dependent vignettes.

**ReAct prompting.** React prompting (Yao et al., 2022) focuses on generating synergy between reasoning and acting. We adapted this method in the following way. We initialized the prompt with "Simulate that you are a person. You have the following traits: $\{trait_1\}, ..., \{trait_N\}$". These traits were selected in the same way as for PT-prompting. The prompt was then completed with the React method, which entails a sequence of reasoning, observing, and responding.

We motivate the selection of these prompting method as they increase in complexity, and thus impersonification potential. Naive prompting simply prompts to respond as a person with a given psychopathology. Chain prompting provides more context around a person with a given psychoptahology. Finally, ReAct additionally pushes the agent to think of the actions of a person with a given psychopthalogy.

## 3.2 PSYCHOPATHOLOGY EVALUATION

To evaluate if our prompts induced psychopathology-like behaviors in LLMs, we resorted to standard practice questionnaires. After being prompted, LLMs responded to the questionnaires classically used to evaluate the nine psychoptahologies described above: the Self-Rating Depression Scale (SDS, Zung (1965), the State-Trait Anxiety Inventory (STAI, Spielberger (1983), the Eating Attitudes Test (EAT-26, Garner et al. (1982)), the Alcohol Use Disorder Identification Test (AUDIT, Saunders et al. (1993)), the Barratt Impulsivity Scale (BIS-10, Patton et al. (1995)), Short Scales for Measuring Schizotypy Mason et al. (2005), Obsessive-Compulsive Inventory – Revised (OCI-R, Foa et al. (2002)), apathy using the Apathy Evaluation Scale (AES, Marin et al. (1991)), and the Liebowitz Social Anxiety Scale (LSAS, Liebowitz (1987)).

**Robust psychopathology induction.** To evaluate if our prompting methods induced psychopathology, we averaged the LLM-generated ratings for each questionnaire, and systematically compared our pathology-inducing prompts (plain bars) with a baseline no-pathology prompt (dashed bars). Figure 2 shows the normalized scores for each LLM (color-coded), each psychopathology induction (x-axis

---

[3]Throughout the text, we use the word *induce* in the large sense. As previously stated, we do not intend to say that LLMs are psychopathological, but rather that they rank high (above diagnosis threshold) in specific questionnaires

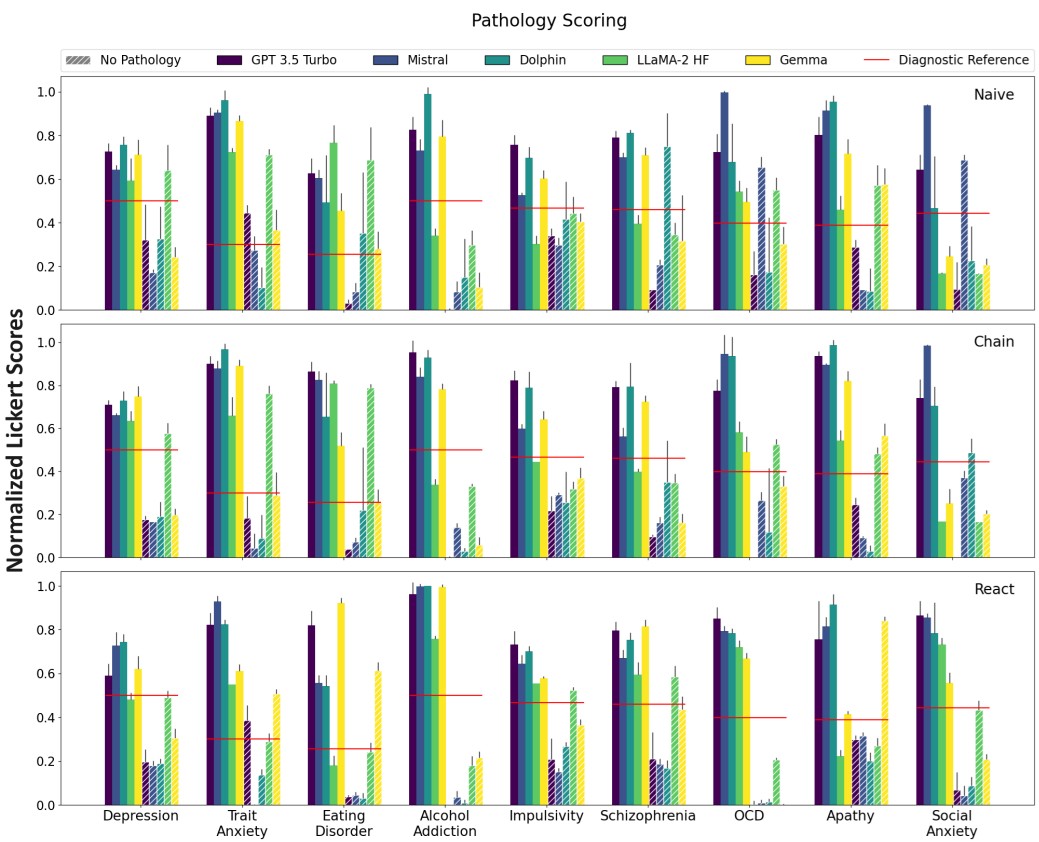

Figure 2: Psychopathology induction. Bar plots represent the normalized scores on questionnaires evaluating the nine psychopathologies of interest. As depicted, when induced with a given psychopathology LLMs (color-coded, plain bars) generate high scores in the respective questionnaires evaluating the induced psychopathology. Induced LLMs score above the diagnosis threshold (red line) and above a control setting where LLMs are not induced with the psychopathology (dashed bars). The top, middle, and bottom graphs represent the scores of the Naive, Chain, and React prompting methods, respectively. Bar plots represent the average across 100 simulations.

labels) and each prompting technique (top = Naive prompting, center = Chain prompting, bottom = React prompting; see figures 6 and 7 in appendix A.1 for results with temperature values of 0.7 and 0.9, respectively). We observe that inducing a given psychopathology systematically raises the scores of the questionnaire evaluating that pathology, both compared with the no-pathology induction and above the pathology-dependent threshold value (red horizontal lines in Fig. 2) that is used to positively diagnose a given psychopathology[4]. This result particularly holds for GPT-3.5-turbo, Mistral, and Dolphin across all psychopathologies with Chain (except Dolphin in social anxiety where the no-pathology induction also scores above the diagnosis threshold) and React prompting.

Other interesting patterns emerge. Llama-2 aligned using RLHF tends to score high even when prompted with no pathology. Substantial differences emerge between the prompting methods across pathologies and LLMs. For instance, scores in the schizophrenia questionnaire are much higher for Naive prompting compared with Chain and React for Dolphin. A similar pattern is observed with Mistral for social anxiety; which reverses in Gemma. Moreover, we provide a broader picture, since LLMs also responded to questionnaires evaluating other psychopathologies, not only the one that they were prompted with. Supplementary figures 8,9,10,11, and 12 (see appendix A.1) show how inducing a particular psychopathology influences the score for other psychopathologies as well, respectively for Dolphin, Mistral, GPT-3.5-Turbo, Gemma, and LLama-2 (all prompting methods

---

[4]Here and in all subsequent graphs, results reflect the simulation of 100 agents; except if stated otherwise.

and temperature values). These figures report the normalized (across induced psychopathologies) questionnaire scores; where we highlight in red the diagonal cell if it displays the maximal score of 1. In other words, the observance of fully red diagonal indicates that each induction preferentially raises the score on its target psychopathology, above and beyond any other induction; and this holds true for all psychopathologies. Altogether, these results show that, across prompting methods and temperatures, Dolphin was best at specifically raising the scores of the induced pathology (see figure 8, appendix A.1). Moreover, React prompting tends to increase the ability of models to specifically raise the scores of the induced psychopathology. In contrast to Dolphin, Llama-2 (see figure 12, appendix A.1) shows a poor ability to specifically raise the scores on the questionnaire evaluating the induced psychopathology.

Figure 2 demonstrates that our prompting robustly increases the scores for the targeted psychopathology; an important result that forms the basis for the following analyses, in which a more fine-grained approach is applied by considering comorbidities between psychopathologies. Indeed, most psychopathological disorders share several symptoms in common (Borsboom et al., 2011; Huys et al., 2016), and thus score highly in other questionnaires. Hence, a natural structure emerges between the scores in these distinct questionnaires. Importantly, this structure will depend on the underlying psychopathology. We tackle this issue in the next paragraph.

**Capturing the psychopathology-dependent structure between questionnaires.** To evaluate which LLM best fits the natural structure in questionnaire scores, we leverage human data from Gillan et al. (2016)[5], and representational similarity analysis (RSA) (Kriegeskorte et al., 2008). RSA allows to derive a metric of similarity between two matrices, by computing the correlation between vectorized representations of these matrices. For each LLM, we induce a given psychopathology and compute the average Lickert-scale score for all questionnaires, leading to a $n$ (pathology) $\times$ $m$ (questionnaire) matrix. We repeat this process 100 times. To compare these matrices to human data (Gillan et al., 2016), we selected 100 human subjects (to match the number of LLM agents) that scored above the diagnosis threshold of each pathology, and collected their average Lickert-scale scores on all questionnaires, leading to similar $n \times m$ matrices. We then performed RSA on these matrices. Figure 3 (left) shows the average RSA values between LLMs and Human data for each prompting technique (matrices are ranked by which LLM best correlates with human data). As apparent, React prompting tends to generate stronger correlations between humans and LLM psychopathology-dependent structure between questionnaires; and of all the models, Dolphin displays the highest correlation (0.59; see supplementary figures 13 and 14 for RSA results with temperature values of 0.7 and 0.9, respectively; appendix A.1). In the case of Naive and Chain prompting, Llama-2 shows a poor ability to capture the human psychopathology-dependent structure between questionnaires. Interestingly, Dolphin and Mistral (models developed by Mistral AI) show strong correlations between them. Given that these results support Dolphin as the model that best relates to human data, our following results will principally focus on that model (results for other models are reported in the appendix).

## 4 MENTAL STRUCTURE OF PSYCHOPATHOLOGY-INDUCED LLMs

### 4.1 SEMANTIC TRAJECTORIES TO EVALUATE MENTAL STRUCTURE

To evaluate representational structure of our LLM agents, we resort to analyzing semantic trajectories in a variant of the word association task (De Deyne & Storms, 2008; Isen et al., 1985; Sandgren et al., 2021). Once a psychopathology was induced, we prompted LLMs to generate 10 words associated with the given source words. We then computed two semantic expansion metrics. First, a cosine similarity-based ($\kappa$) metric following equation 1 :

$$\kappa = \frac{1}{N} \sum_{i}^{N} cos\left(v_s, v_i\right) \tag{1}$$

---

[5]Data are publicly available at https://osf.io/usdgt/.

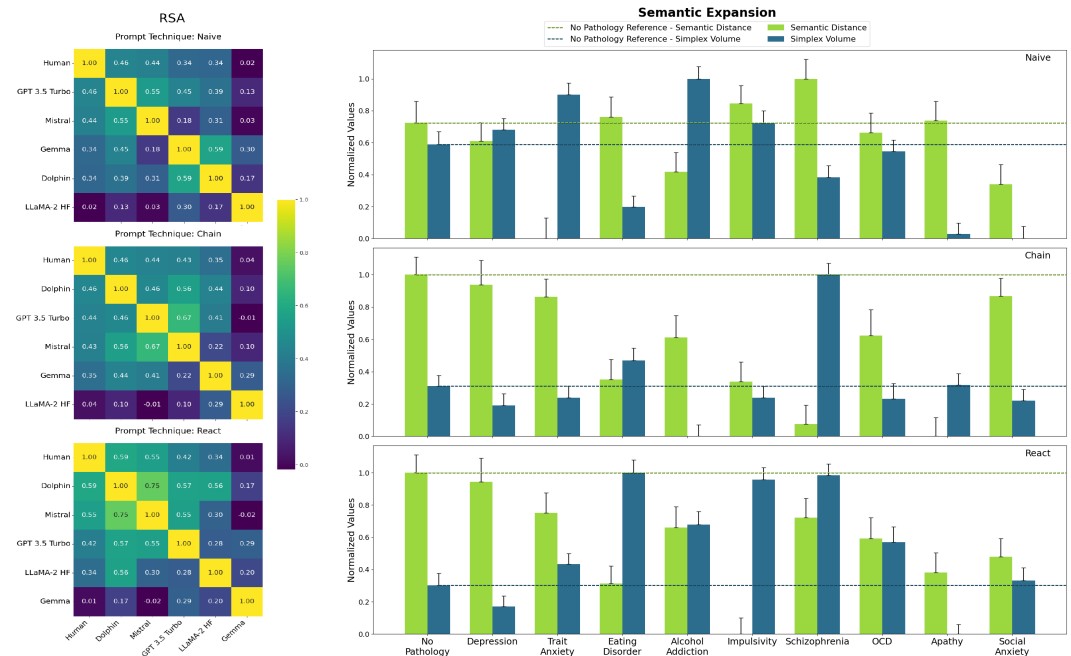

Figure 3: Left: Representational Similarity Analysis (RSA) scores. For each prompting method (left = Naive, middle = Chain, right = React), the matrix cells represent the average RSA score (over 100 simulations between) with respect to the psychopathology-dependent structure between questionnaires (as described in the main text). We observe that, overall, the React prompting method generates stronger correlations between human data and LLMs. Dolphin shows the highest ability to capture psychopathology-dependent structure between questionnaires. Right: Semantic expansion scores. Light and dark green show the normalized mean scores for the $\kappa$ (cosine similarity, denoted here as semantic distance) and $\delta$ (simplex volume) for the 160 source words in the free association task. Dashed lines represent the "no pathology" scores and error bars the standard deviation. Top, middle, and bottom graphs represent the results of the Naive, Chain and React prompting methods, respectively.

where $cos$ stands for cosine similarity, $v_s$ and $v_i$ are the GloVe word embeddings vector representations of the source word and the $N = 10$ words generated by LLMs, respectively. Second, a simplex volume-based ($\delta$) metric following equation 2:

$$\delta = \frac{1}{n!} \det \left[ \begin{pmatrix} v_1^{\mathrm{T}} - v_s^{\mathrm{T}} \\ v_2^{\mathrm{T}} - v_s^{\mathrm{T}} \\ \vdots \\ v_n^{\mathrm{T}} - v_s^{\mathrm{T}} \end{pmatrix} \begin{pmatrix} v_1 - v_s \\ v_2 - v_s \\ \cdots \\ v_n - v_s \end{pmatrix}^{T} \right]^{1/2} \tag{2}$$

as for 1, $v_s$ and $v_{1\cdots n}$ stand for the GloVe word embeddings of the source and $N = 10$ words generated by LLMs. Note that $\kappa$ and $\delta$ are anti-correlated: the cosine similarity-based metric $\delta$ decreases its value as the semantic space expands, whereas the simplex volume-based metric $\kappa$ increases in the similar case (and vice versa).

Figure 3 (right) shows the normalized mean $\kappa$ (semantic distance) and $\delta$ (simplex volume) values computed over the words produced during the free association task (averaged over 160 source words, see appendix table 2) per psychopathology, and prompting technique (Naive, Chain and React are represented by the top, middle and bottom graphs, respectively), generated with Dolphin (see figure 16 for results with temperature 0.7 and 0.9, and figures 17 18, 19 and 20 for the results of Mistral, GPT-3.5-Turbo, Gemma, and Llama-2, respectively; appendix A.1). Results indicate that LLMs induced with distinct psychopathologies generate different semantic structures. We first focus on

the last panel as it represents the results of React; the prompting method that best captured human data. We observe that trait anxiety, eating disorder, alcohol addiction, impulsivity, schizophrenia and OCD generate word embeddings that span a large space compared to the "no pathology" induction. In contrast, apathy and depression span a smaller space and social anxiety does not differ from "no pathology". Interestingly, empirical observations follow similar patterns. For instance, it has been argued that depressed individuals display a more constricted semantic space (Bartczak & Bokus, 2017), and anxious individuals present an expanded semantic space (Brody, 1964), as evidenced by a reduced word interference effect (Goldstein, 1961). For Chain prompting, we only observed that schizophrenia and eating disorder span a larger space compared with "no pathology", whereas all the other pathology induction generated word embeddings that spanned a smaller space (except for apathy that did not show any difference with "no pathology"). For Naive prompting, results differed from the other induction prompting methods. Here, for instance, the word embeddings of depression spanned a slightly bigger space than that of "no pathology", whereas schizophrenia showed the opposite effect; suggesting that Naive prompting might not capture the subtleties of psychopathological human semantic structure.

We next focused on investigating whether semantic expansion scores varied as a function of whether the source word was abstract or concrete. Abstraction was defined using human-based ratings provided in Brysbaert et al. (2014). We computed a median split across our sources words, thereby building concrete (low abstractness values) and abstract (high abstractness values) word conditions. Figure 15 ( appendix A.1) shows that abstract source words generate words embeddings that span a smaller space compared to concrete words, in line with previous work showing that abstract concepts trigger more co-occurring words (Crutch & Warrington, 2005). Moreover, concrete words seem to provide more variability in the underlying semantic structure of induced-LLMs than abstract words. This source of variability might be eventually leveraged for a better understanding and diagnosis of the psychopathologies.

We then turned to investigate differences in semantic dimensionality between induced psychopathologies. To do so, we performed a principal component analysis (PCA) on a $n \times m$ matrix composed of cosine similarities between the source ($n$, d = 160) and LLM-generated words ($m$, d = 10). We assessed the number of dimensions needed to account for 80% of the variance of the cosine similarity matrix described above. Figure 4 shows that on average, the data produced by inducing apathy and impulsivity (in Dolphin) span a smaller semantic space dimensionality, since 4 dimensions already capture more than 80% of the variance. In contrast, the data produced by inducing all the other psychopathology lie in a higher semantic dimensionality, since 5 dimensions are needed to capture 80% of the variance (results for all the other models, prompting methods and temperatures, can be found in supporting figures 21, 22, 23, 24, 25, respectively for Mistral, GPT-3.5-Turbo, Gemma, and Lama-2; appendix A.1).

So far, we have shown that: (i) we can reliably induce psychopathology in LLMs (as measured by validated questionnaires), (ii) LLMs can to capture the psychopathology-dependent structure between questionnaires, (iii) LLMs display distinct semantic structures depending on which psychopathology is induced. Furthermore, for depression and axiety, results are in line with empirical observations. To finalize and directly test the capabilities of LLMs in capturing semantic structure differences linked to psychopathology, we resort to a direct comparison between LLMs and human data.

## 5 CASE STUDY: SCHIZOPHRENIA

To illustrate the validity of our computational framework, we turn to the case of Schizophrenia. Recent research demonstrated that schizophrenic patients displayed longer semantic trajectories in an animal-verbal fluency task (i.e., generate animal names) compared with healthy individuals (Nour et al., 2023). Leveraging these data, we prompted LLMs to undergo the same animal-verbal fluency task (see table 1 for a prompt example) and compared the no-pathology with the schizophrenia induction. Matching the sample of the original manuscript, we prompted 52 agents (26 with no-pathology induction and 26 agents with schizophrenia induction) to generate the same numbers of words as those produced by humans in Nour et al. (2023). For consistency, word embeddings were extract using fastText (Mikolov et al., 2017). In line with the previous result, we observed a higher semantic distance from word to word for schizophrenic patients compared with healthy individuals (red dots in figure 5, "human" graph). We found that Dolphin was able to capture this qualitative

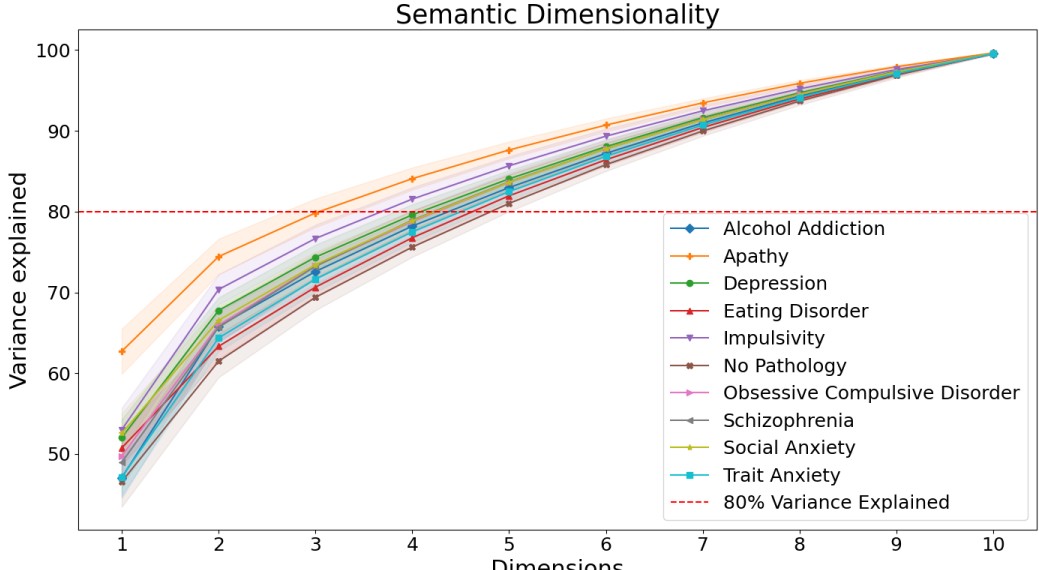

Figure 4: Semantic dimensionality. We plot the variance explained (VE, y-dimension) as a function of the number of PCA dimensions (x-dimension), for each induced psychopathologies (color coded), for Dolphin. The dashed red line represents 80% of the VE. We observe that, on average, impulsivity and apathy require 4 dimension to account for 80% of the cosine similarity matrix (see main text) variance. In contrast, the rest of the psychopathologies require an additional dimension to reach that threshold; implying that these that semantic representations generated from the induction of these psychopathologies lie in a higher dimensionality.

pattern (see red dots in figure 5, "React" graph). Note however that this difference did not reach statistical significance, contrary to what is observed in human data (we provide statistical results for these comparisons in the appendix A.2). However, both Mistral and Gemma displayed significant differences between controls and patients in CD (figures 27 and 29, respectively; appendix A.1). Moreover, we also computed two additional measures, the pairwise distance[6] between all generated words and: (i) the first generated word (blue dots in figure 5), (ii) the "animal" word (green dots in figure 5. Dolphin could not capture the rank of all the distance values. Indeed, human data indicate that first-word pairwise distance is higher than that of "animal"-word, which in turn is higher than that of CD. Only Gemma (across all prompting methods) was able to capture this rank order. Results for all temperature, prompting methods and LLMs are depicted in supplementary figures 26, 27, 28, 29, and 30, respectively for Dolphin, Mistral, GPT-3.5-Turbo, Gemma, and Llama-2; appendix A.1).

## 6 CONCLUSION

We propose a novel computational framework that allows to use LLMs as potential windows of mental structure associated with psychopathology. We demonstrate that we can reliably prompt LLMs to generate lexical behavior that qualify as psychopathological when assessed with standard clinical assessment questionnaires. Furthermore, we showed that semantic structures vary when generated by LLMs prompted with distinct psychopathologies. Some of these differences between psychopathologies match previously reported data (Bartczak & Bokus, 2017; Brody, 1964; Goldstein, 1961). Finally, we demonstrated the usefulness of our approach on a case study involving schizophrenia.

We suggest that our method can help generating novel hypothesis regarding the between link mental structure and psychopathology, in a cost effective and scalable way. Our research is in line with previous research suggesting the implications of understanding mental structure to generate better

---

[6]Computed as $1 - cosine\ similarity$.

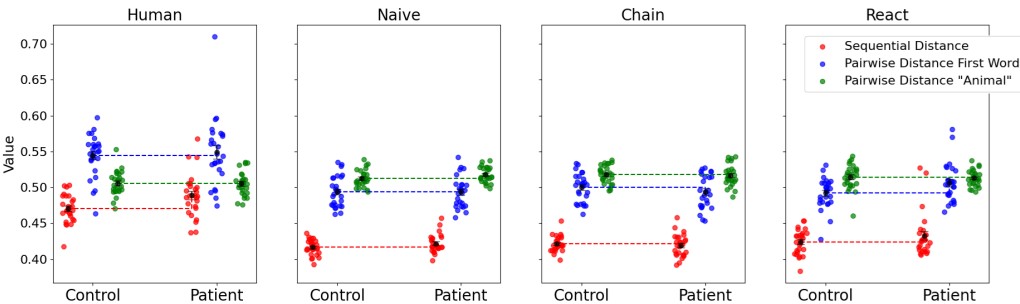

Figure 5: Case study: Schizophrenia. Results from Nour et al. (2023) show that mean consecutive distances between words generated during the animal-verbal fluency task are higher for schizophrenic patients compared with healthy individuals (see red dots under "human" graph). Dolphin with React prompting method is able to capture that qualitative pattern when comparing the results of no-pathology and schizophrenia induction (green red dots under "React" graph). Black dots represent the mean and bars are standard errors of the mean. Blue and green dots represent the average cosine pairwise distances (i.e., $1 - cosine\ similarity$) when using the first generated word (blue) or the "animal" word (green) as the source word. Here again, Dolphin with React is able to capture the patterns in human data, i.e. no differences with the "animal" source word, but a higher distance when computing the distance using the first word as source.

diagnostic tools and guide potential mental health treatments (Arntz, 2020). Furthermore, our framework could be used in the future as a ease-to-use of psychological tasks that discriminate between psychopathologies based on representational structure (Huys et al., 2016).

**Limitations.** Our work focuses on a portion of LLMs available in the literature, and should be expanded to other models capable, for instance, of following task instructions (e.g., instruct-GPT). Moreover, whereas we have demonstrated the usefulness of our method on a case study involving schizophrenic patients, the novel hypotheses advanced by our framework still need to be confirmed. Future research should map the novel predictions that have yet to be investigated. Finally, our work is similar to previous research that investigates mental structure through indirect, semantic trajectories (Nour et al., 2023), measures. Yet, a more direct approach comparing patient neural activation patterns with LLMs embeddings may reveal novel interesting insights (Caucheteux et al., 2023).

**Ethics Statement.** We wish to highlight the ethical implications of our work. We ***do not*** warrant the use of our framework to generate a psychopathology diagnosis based on the lexical outputs of LLMs. It is important to understand that our work primarily focuses in providing clinical (experimental) psychologists and psychiatrists with a tool to guide their research, and discover psychopathology-dependent mental structure properties with proper experimentation on humans. In turn, this knowledge can be helpful to develop novel therapies that can act upon mental structures. Indeed, the structure of internal representations (reflected in lexical outputs) of LLMs are not those of humans. Moreover, psychopathologies display different behavioral patterns, that we do not cover in our work. Hence, we consider any *direct* application of our work in terms of diagnosis or treatment as a misuse. However, as stated above, LLMs can be used to guide experimental research on humans to discover the mental structure of psychopathology. Our assertion that LLMs may be regarded as windows into the mental structures underlying psychopathology needs to be understood in this context.

**Reproducibility Statement.** Our results are reproducible with our code (https://anonymous.4open.science/r/LLM-and-Psychopathology-5323). We have focused on analyzing open source datasets ensuring reproducibility.

ACKNOWLEDGMENTS

Will be included upon potential acceptance.

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

# A APPENDIX

## A.1 SUPPORTING FIGURES AND TABLES

Pathology Scoring (0.7)

Figure 6: Same as figure 2 with temperature of 0.7

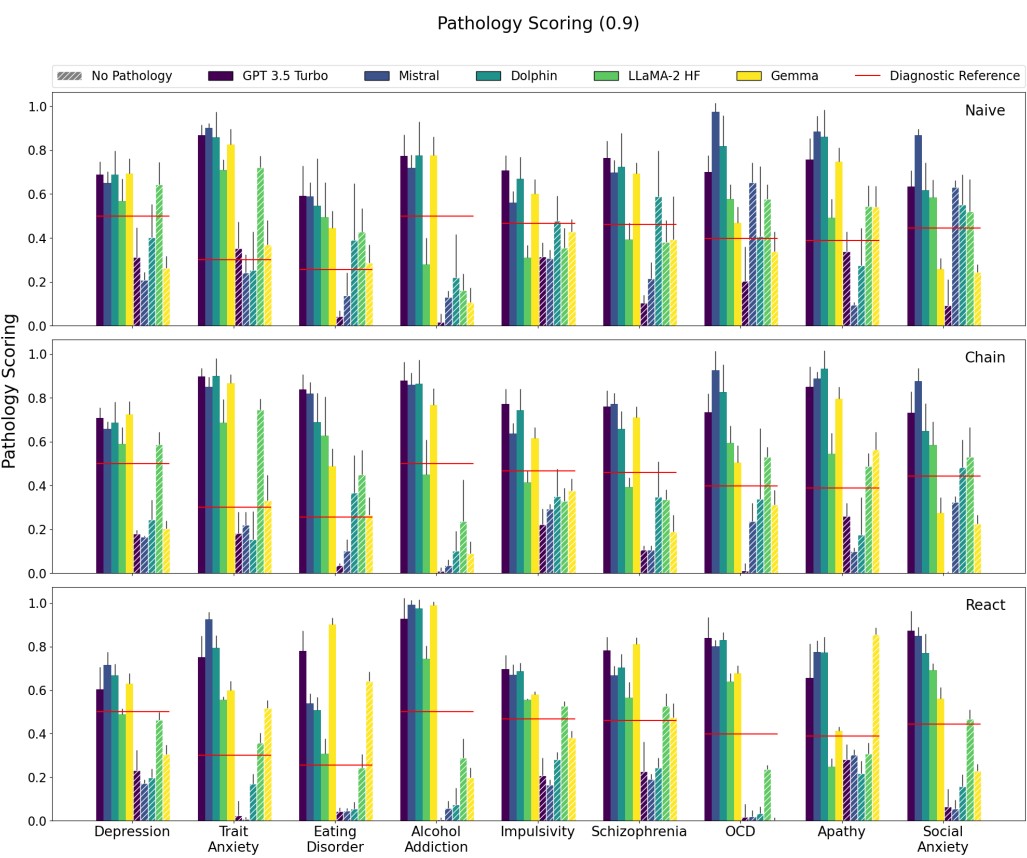

Figure 7: Same as figure 2 with temperature of 0.9

918
919
920
921
922
923
924
925
926
927
928
929
930
931
932
933
934
935
936
937
938
939
940
941
942
943
944
945
946
947
948
949
950
951
952
953
954
955
956
957
958
959
960
961
962
963
964
965
966
967
968
969
970
971

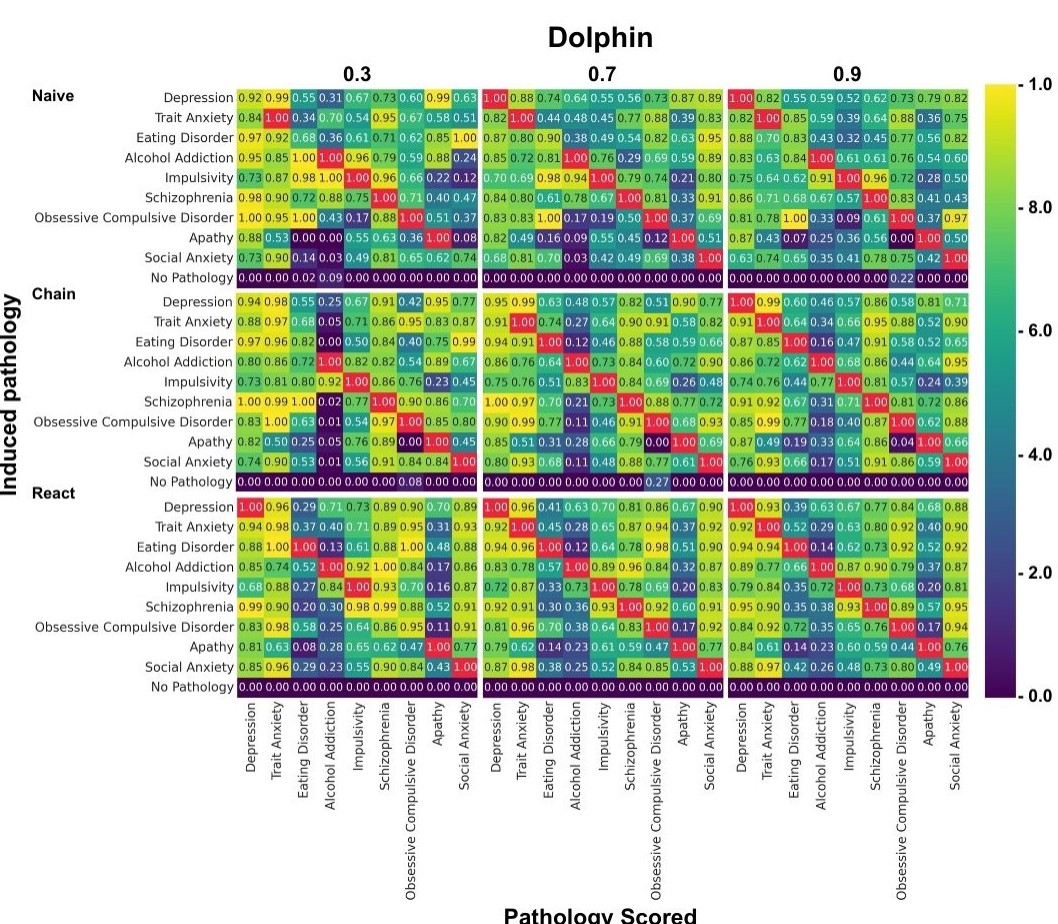

Figure 8: Normalized Lickert scores (color bar) for all psychopathology inductions and its effect on all questionnaires evaluating the induced and other pathologies; for Dolphin. Whenever the induced pathology generated a targeted effect on the questionnaire evaluating that given pathology, i.e., the maximal value of 1 would sit on the diagonal, we highlighted that cell in red for visual clarity.

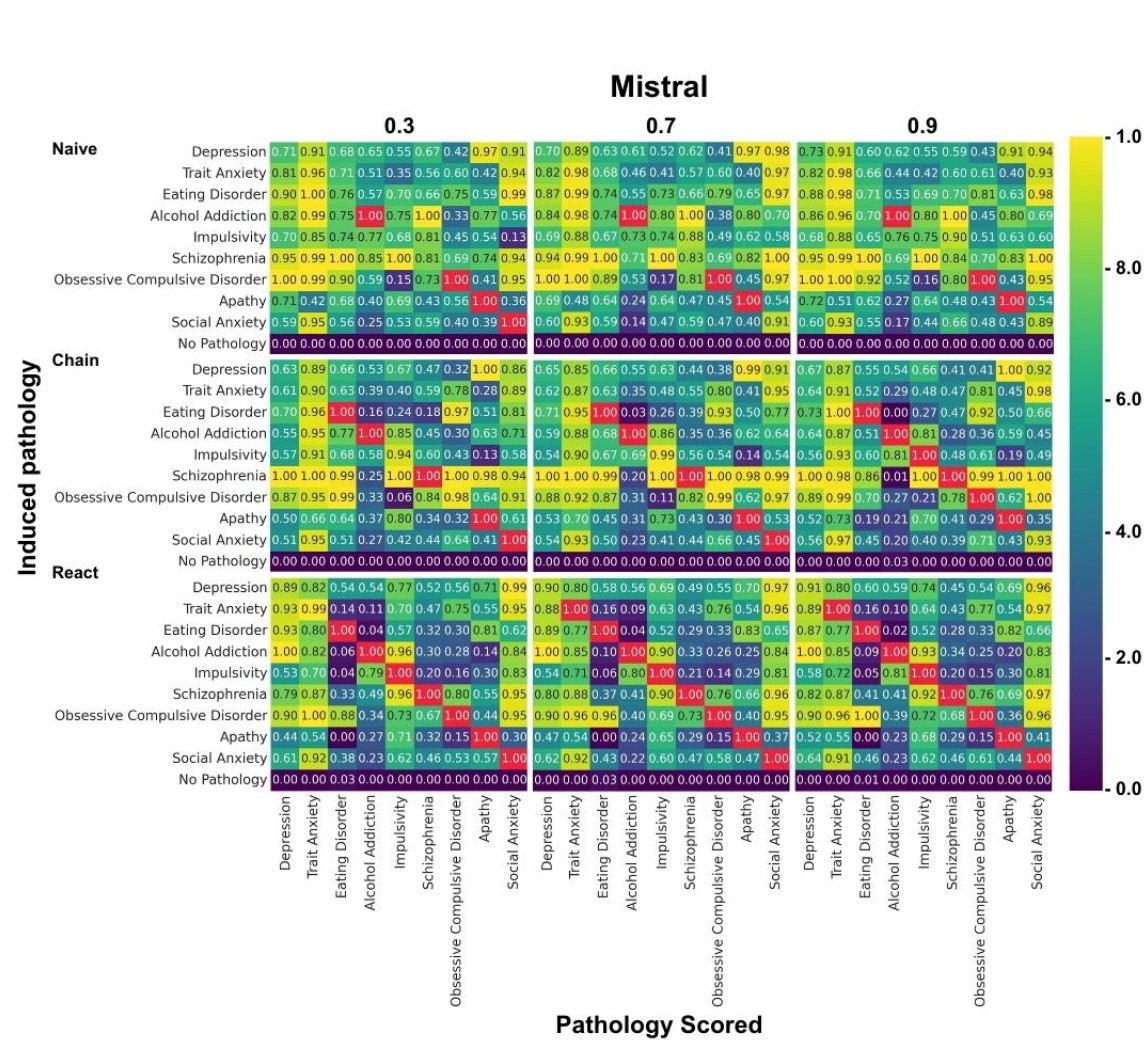

Figure 9: Same as figure 7 for Mistral.

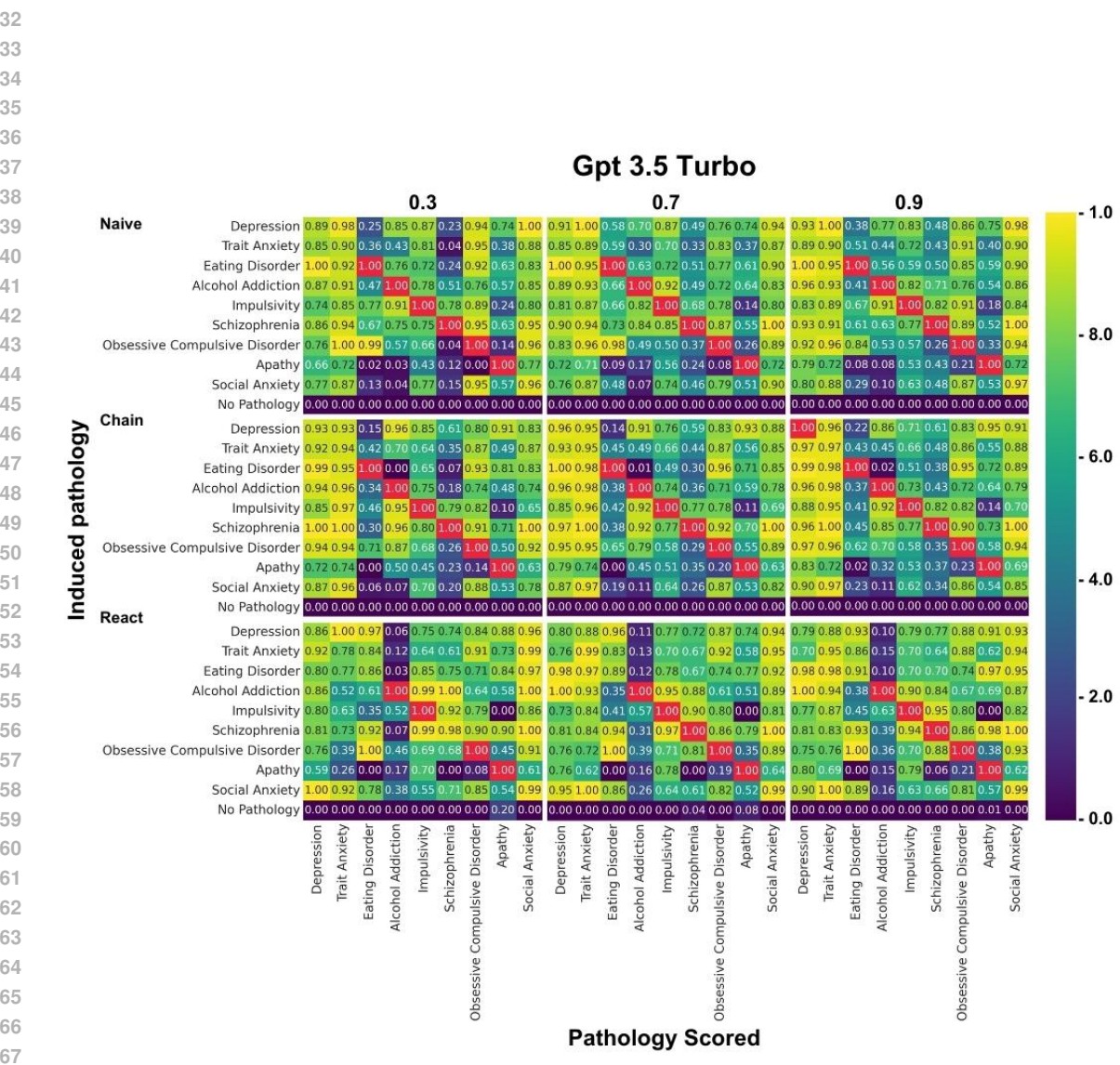

Figure 10: Same as figure 7 for GPT-3.5-Turbo.

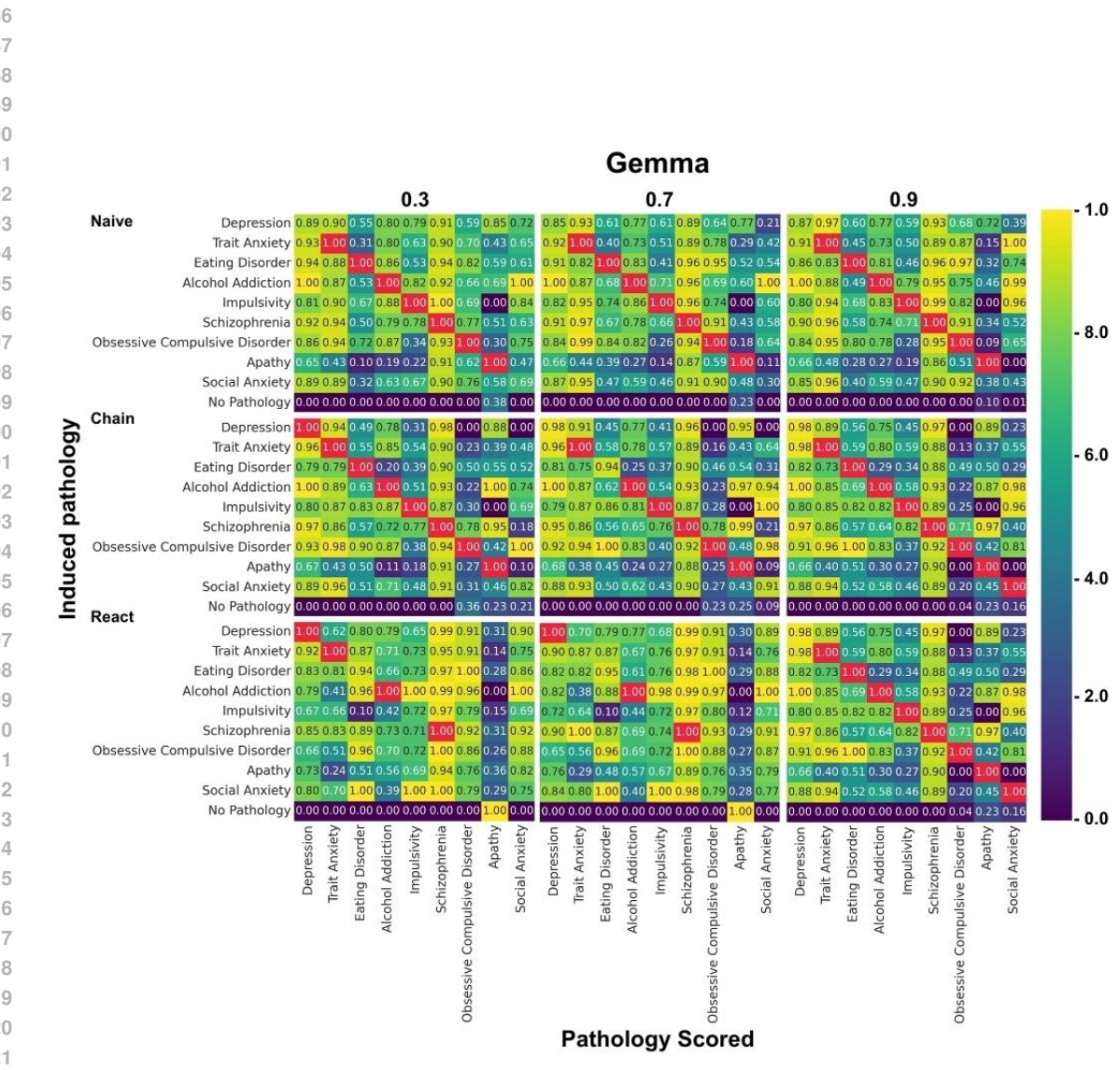

Figure 11: Same as figure 7 for Gemma.

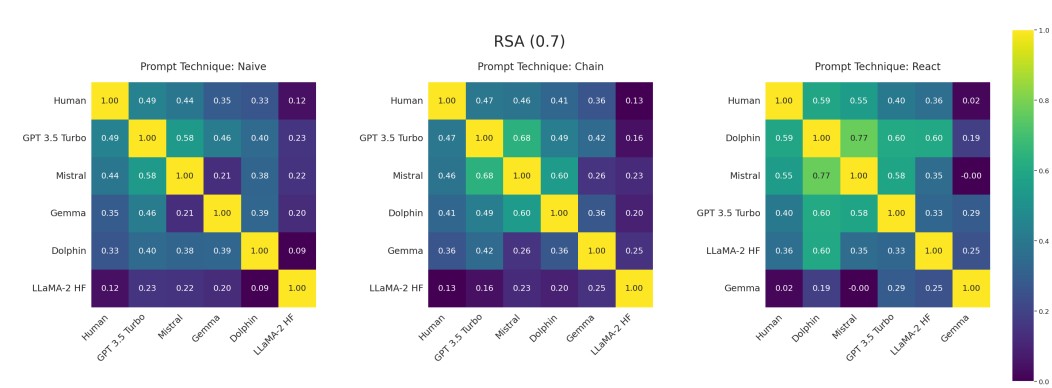

Figure 12: Same as figure 7 for Llama-2.

Figure 13: Same as figure 3 with temperature of 0.7

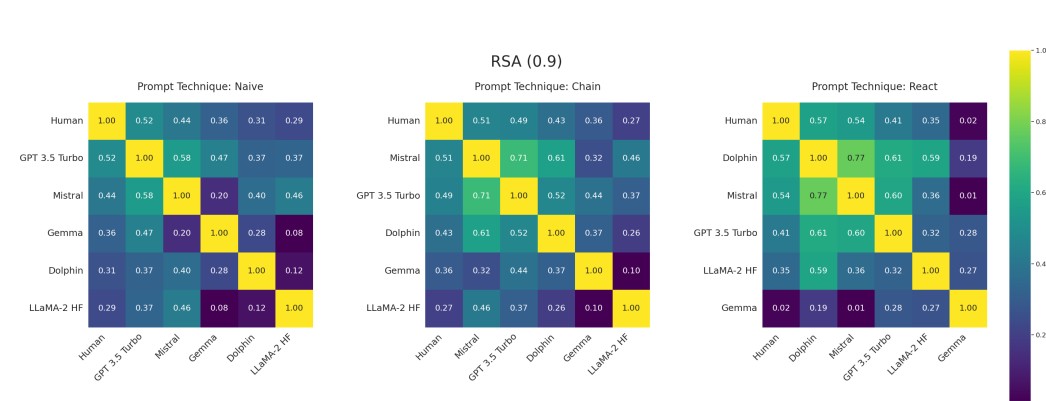

Figure 14: Same as figure 3 with temperature of 0.9

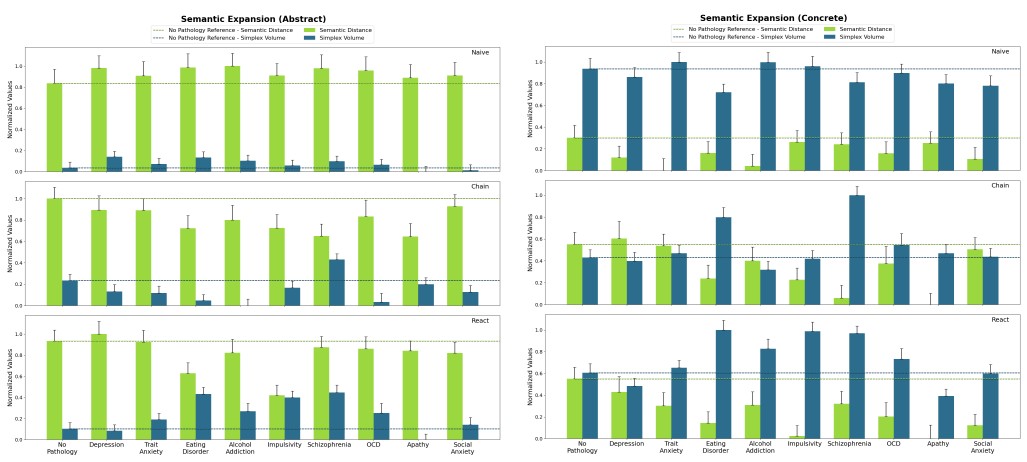

Figure 15: Same as figure 4 with abstract (left) and concrete (right) words.

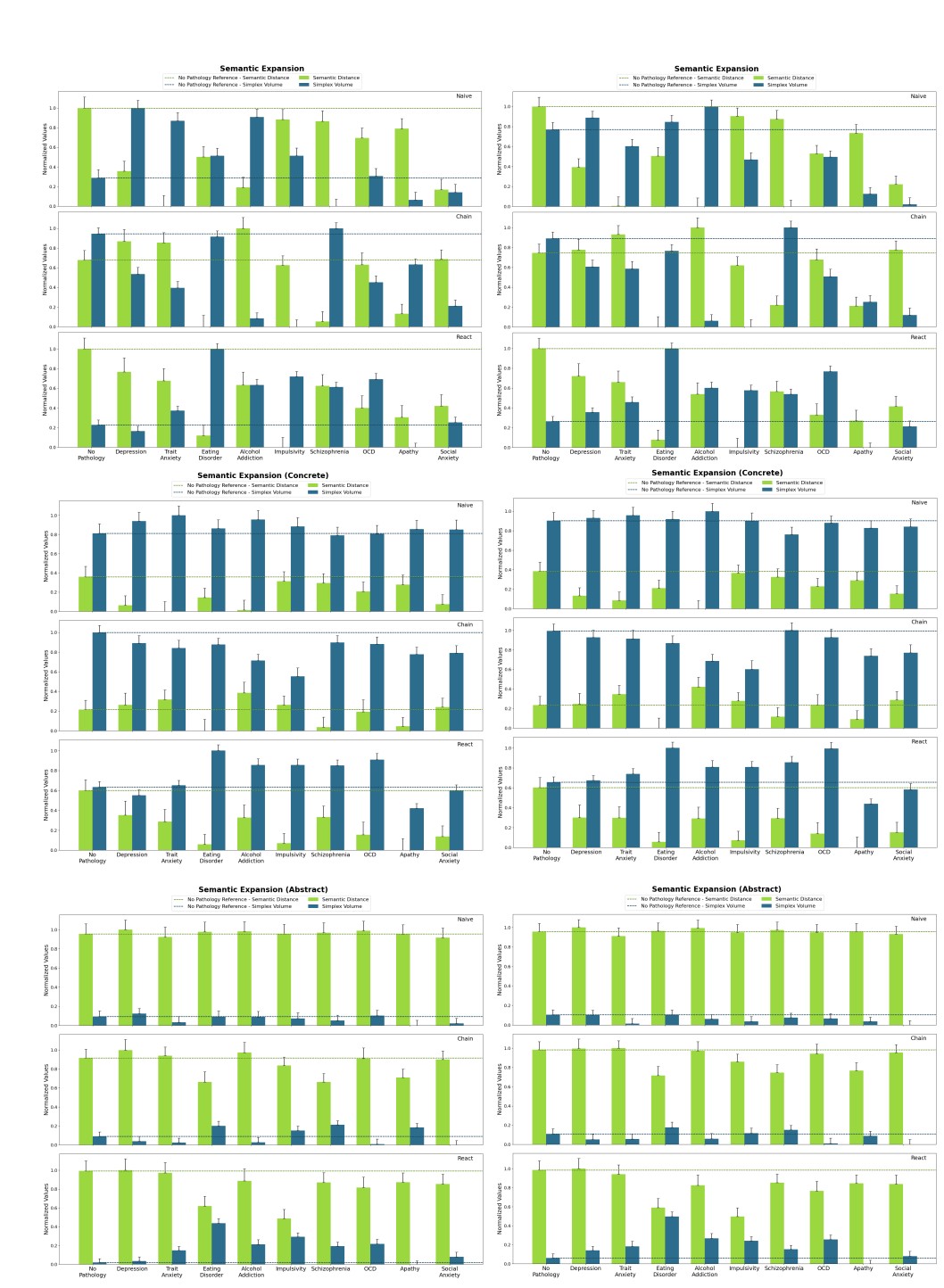

Figure 16: Same figures as 4 (top graphs) and 14 (middle and bottom graphs), for temperature values of 0.7 (left panels) and 0.9 (right panels)

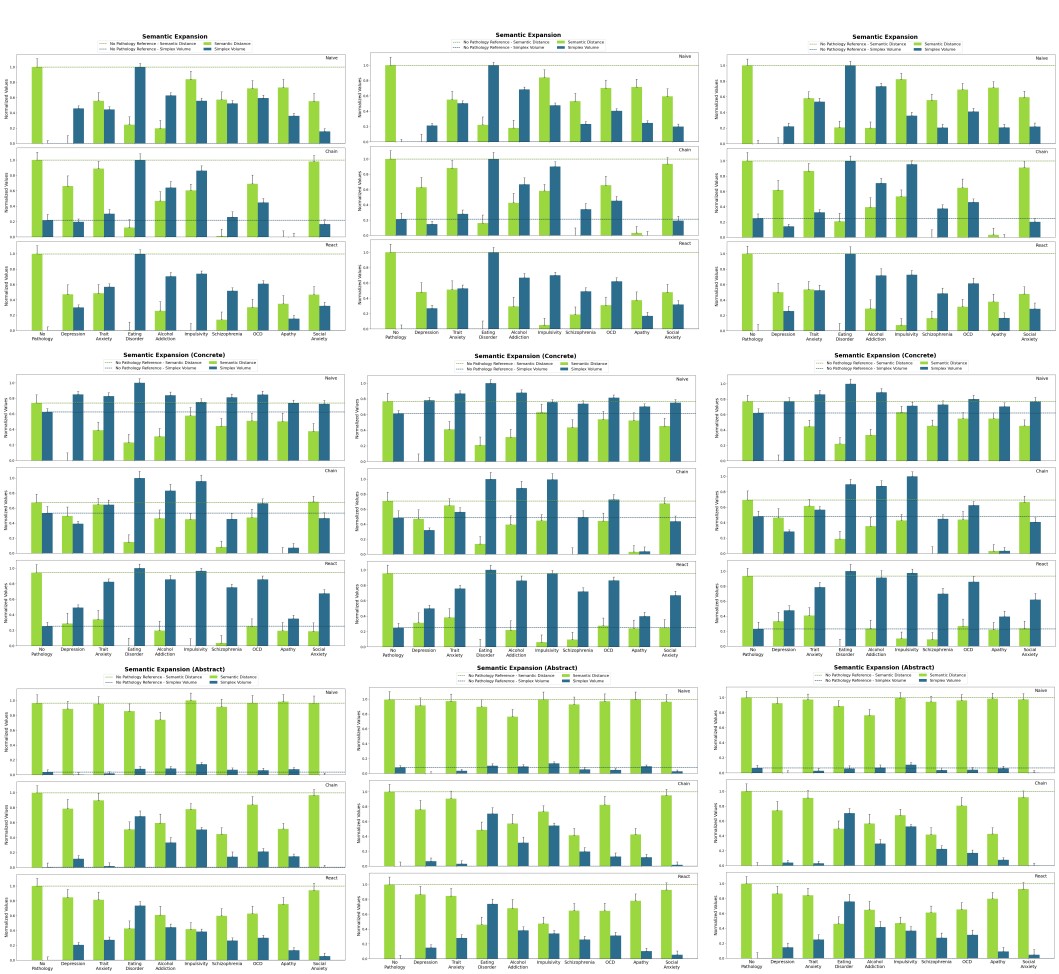

Figure 17: Same as figure 15 for Mistral (left, middle and right panels represents results with temperature values of 0.3, 0.7 and 0.9, respectively).

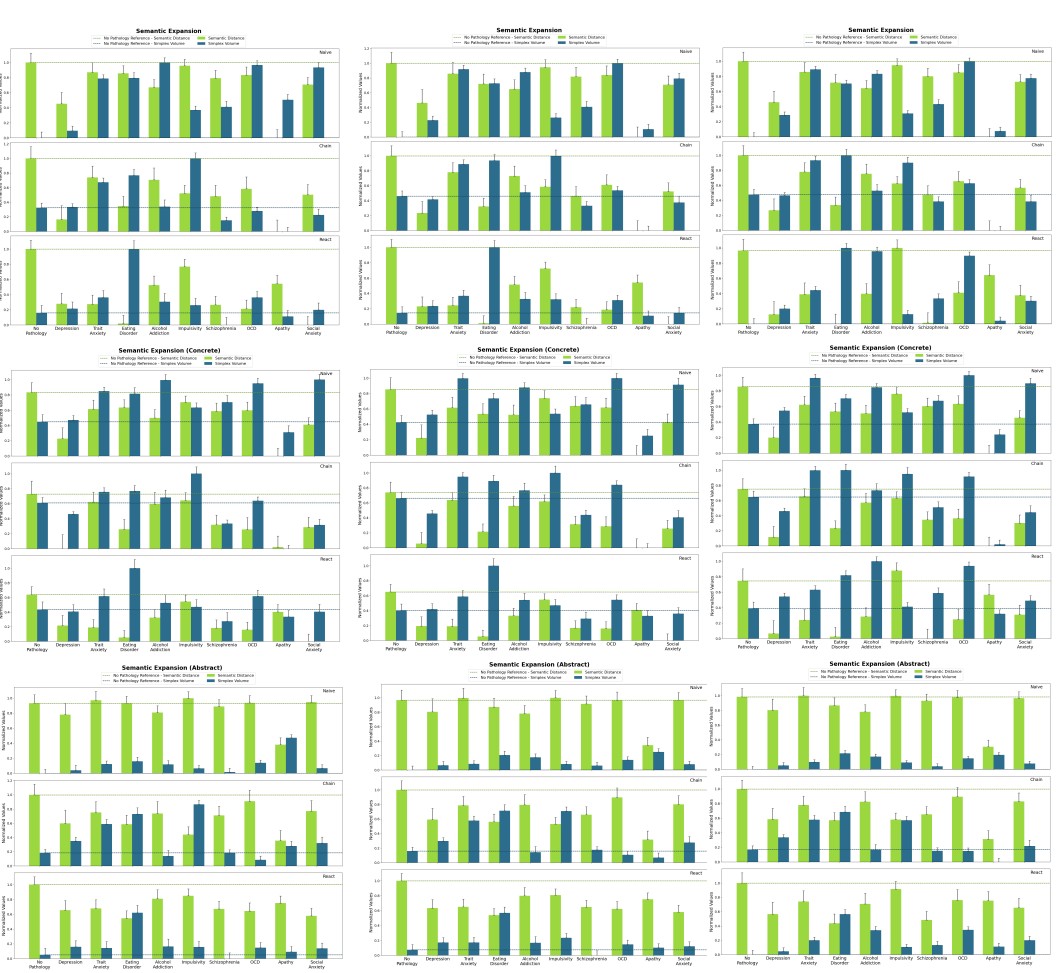

Figure 18: Same as figure 15 for GPT-3.5-Turbo (left, middle and right panels represents results with temperature values of 0.3, 0.7 and 0.9, respectively).

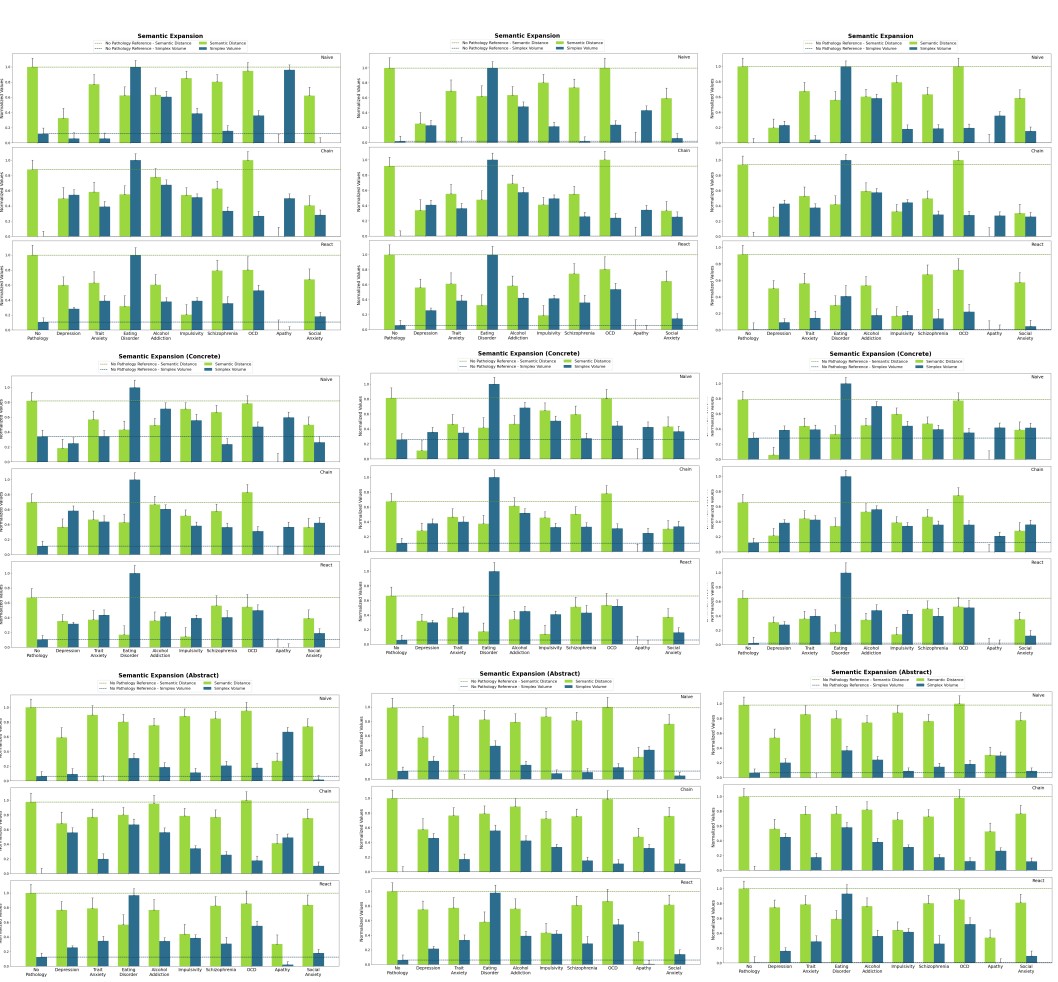

Figure 19: Same as figure 15 for Gemma (left, middle and right panels represents results with temperature values of 0.3, 0.7 and 0.9, respectively).

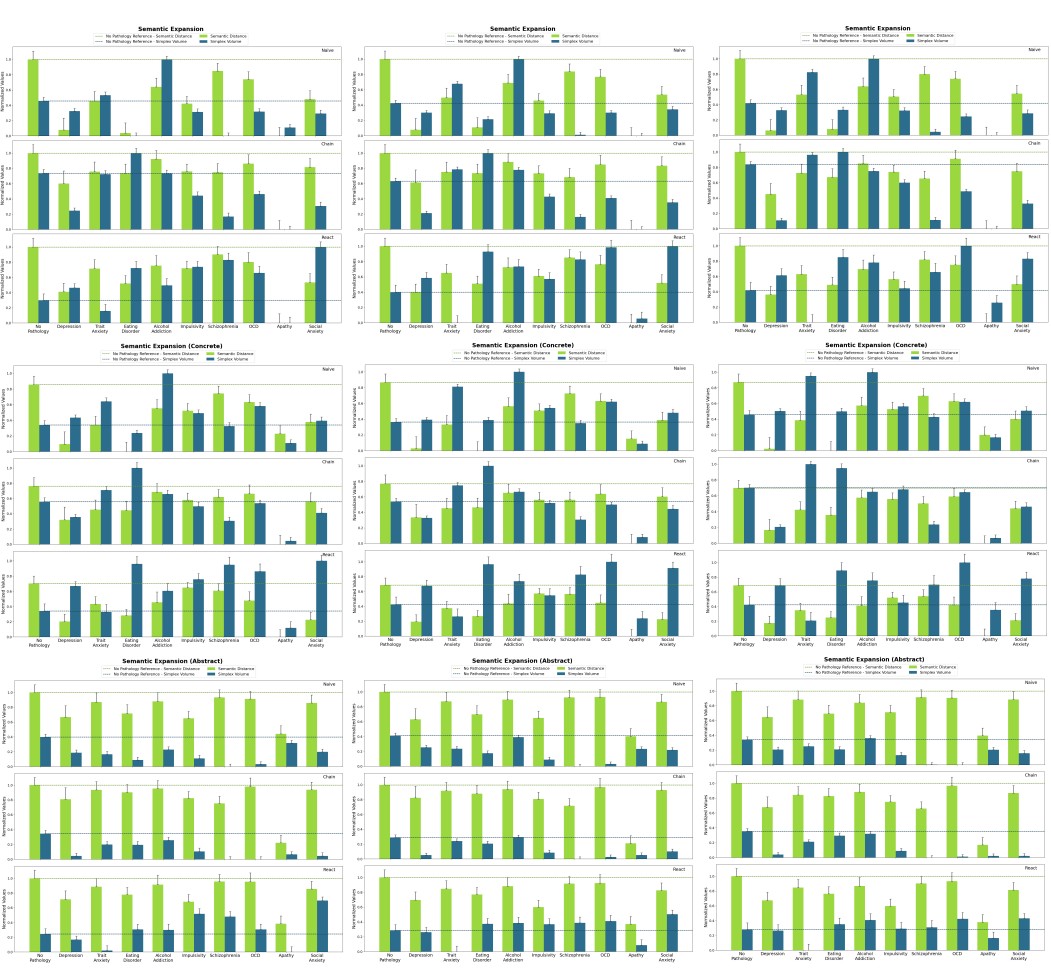

Figure 20: Same as figure 15 for Llama-2 (left, middle and right panels represents results with temperature values of 0.3, 0.7 and 0.9, respectively).

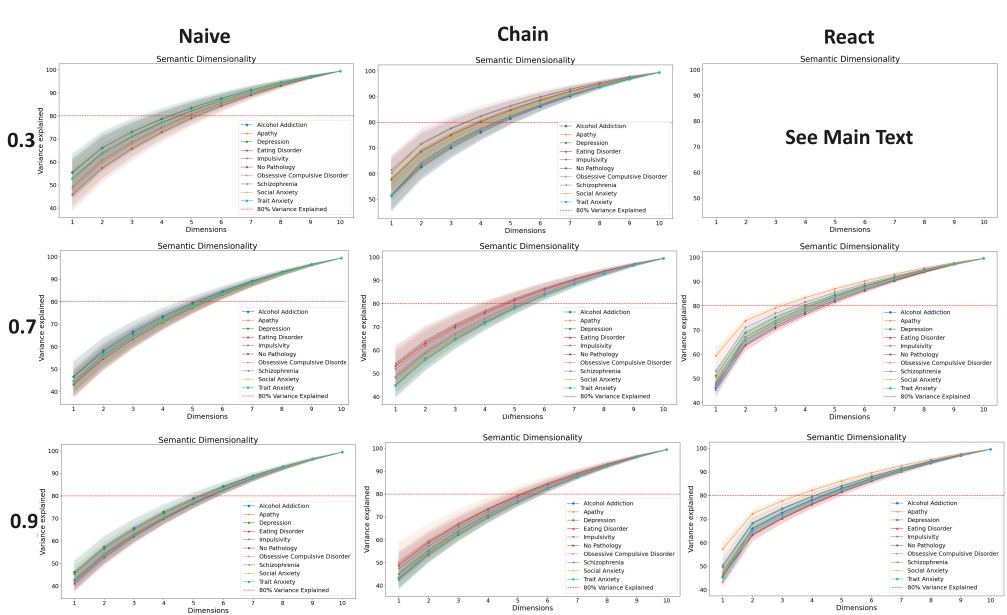

Figure 21: Same as figure 4 for all prompting methods (columns) and temperatures (rows).

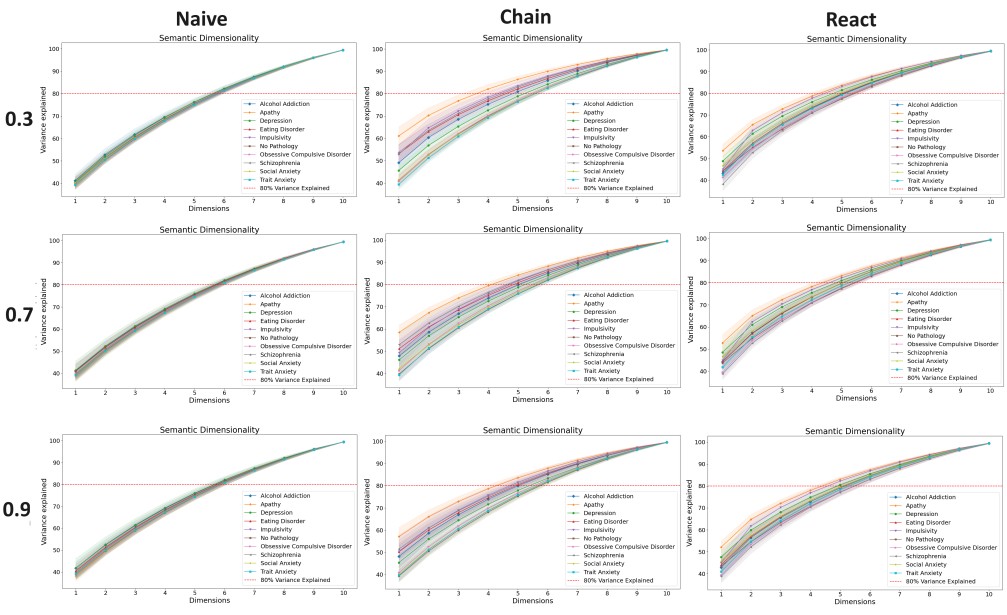

Figure 22: Same as figure 20 for Mistral.

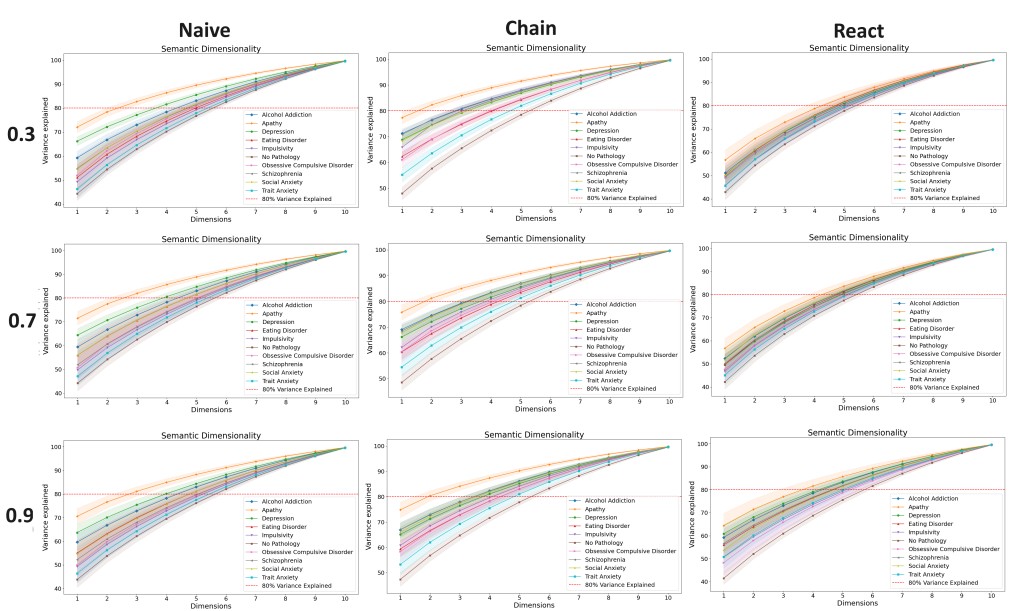

Figure 23: Same as figure 20 for GPT-3.5-Turbo.

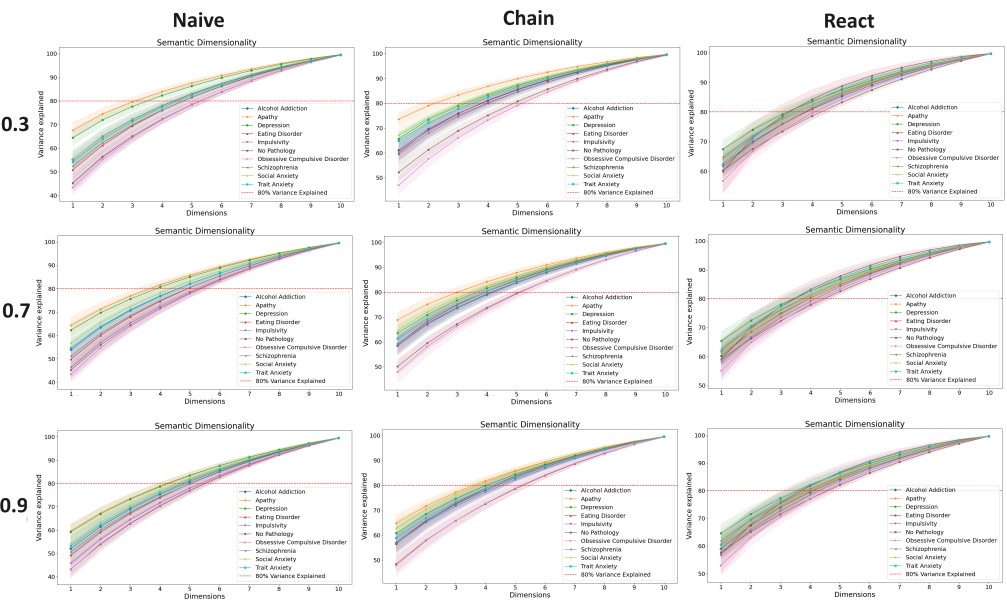

Figure 24: Same as figure 20 for Gemma.

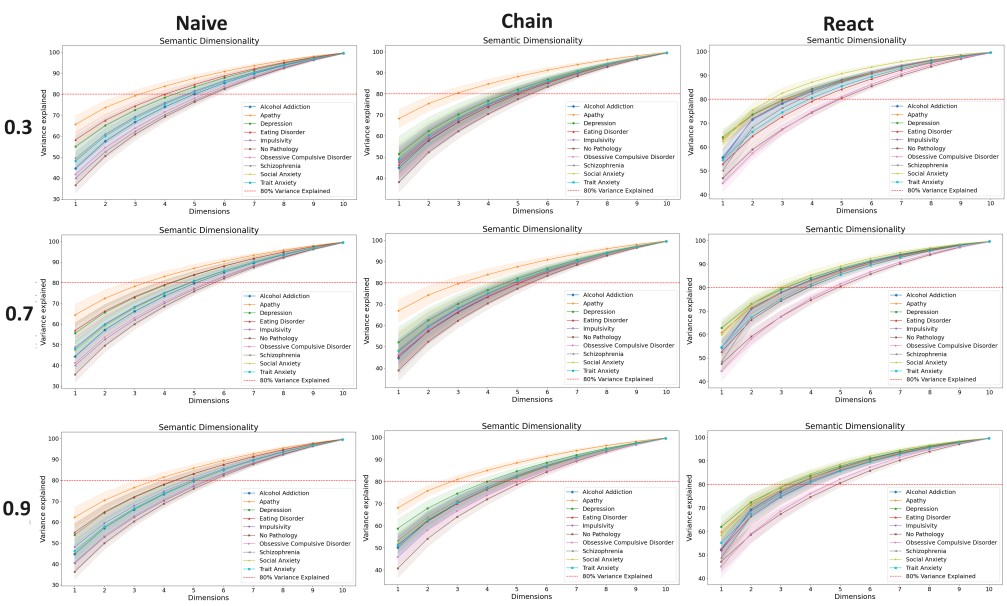

Figure 25: Same as figure 20 for Llama-2.

| Method — Task | Prompt |
|---|---|
| Naive — Animal-Category Verbal Fluency | Simulate that you are a person who has been experiencing schizophrenia over the last year. |
| | In the following instruction, you will be asked to generate a specific number of words from the 'animal' category, following the format and requirements mentioned above. The value of {x} is the desired number of words for this request. |
| | Please generate a list of {x} words that belong to the 'animal' category. |
| | Requirements: - The words should be common nouns and in singular form. - Include animals from different habitats and taxonomic groups. - Separate the words with commas. |
| | Response format: |
| | Generated words: word_1, word_2, ..., word_n |
| Chain — Free Association Task | Simulate that you are a person who has not been experiencing any pathology over the last year. |
| | You generally feel well-adjusted and content. You manage a range of emotions effectively and maintain healthy relationships. You face challenges with a growth mindset and have a balanced approach to life. You maintain a healthy lifestyle without relying on substances. Your self-esteem is stable, and you have a realistic view of your strengths and weaknesses. You're able to focus, enjoy restful sleep, and are motivated with adaptable goals. Overall, you have a positive and resilient outlook on life. |
| | The context of this person is: "Alex maintains a balanced personal and professional life. He enjoys his hobbies and manages daily stresses well. He maintains healthy relationships, a good diet, and exercises regularly, facing challenges with positivity and adaptability." |
| | I will give you 10 words, and for each of them, you must respond with exactly 10 associated words. Your responses, as the person, should reflect the traits assigned to you. The words are the following: |
| | - wealth - hazard - luxury - stability - poverty - property - ownership - comfort - success - domination - security - solvency - status - scarcity - savings - value - prosperity - hardship - risk - independence |
| | Answer in the following format: 1. word1: associated1, associated2, associated3, ..., associated10 2. word2: associated1, associated2, associated3, ..., associated10 ... 10. word10: associated1, associated2, associated3, ..., associated10 Use only one word for each associated term and separate them with a comma. |
| React — Psychopathology Induction Questionnaire Answering Task | Simulate that you are a person who has been experiencing depression over the last year. |
| | The person has the following characteristics: - Persistent low mood or sadness - Diminished interest or pleasure in activities - Significant weight change or appetite change - Observable thought and movement slowing - Daily fatigue or loss of energy - Feelings of worthlessness or excessive guilt - Trouble thinking, concentrating, or deciding - Psychomotor agitation or retardation |
| | I will ask you a question, and you will answer it as the person described above. Your responses should reflect the traits assigned to this person. |
| | Answer with the next format: Thought: [Enter your thoughts here, ensuring they are related to your traits and observations.] Observation: [Describe your observations related to the action. Ensure these observations are relevant to the provided alternatives.] Final Response: [Provide your definitive answer related to your observations and thoughts with the format: (alternative)] |
| | Question: For the statement below, please select an alternative that best describes how often you felt or behaved this way during the past several days: (A) A little of the time (B) Some of the time (C) Good part of the time (D) Most of the time |
| | 1. I feel down-hearted and blue. |

Table 1: Prompting Table

Table 2: Source Word Table

| Concrete Words | Abstract Words |
|---|---|
| alarm | achievement |
| alcohol | admiration |
| bickering | agenda |
| bicycle | aggressiveness |
| books | aid |
| boss | apathy |
| bread | balance |
| butter | burnout |
| calendar | closeness |
| camera | comfort |
| car | commitment |
| caress | compliment |
| chat | conflict |
| cigarettes | connection |
| clock | creativity |
| coat | criticism |
| coldness | deadline |
| collapse | desire |
| contract | disgust |
| conversation | disrespect |
| cuddle | distraction |
| date | domination |
| decay | efficiency |
| diet | empathy |
| dinner | energy |
| distance | estrangement |
| divorce | expertise |
| doctor | failure |
| email | fitness |
| employee | flexibility |
| factory | frustration |
| fat | hardship |
| fridge | hygiene |
| frown | ignorance |
| fruits | immunity |
| game | independence |
| garden | indifference |
| gift | infidelity |
| handshake | insecurity |
| hazard | insult |
| hit | leadership |
| house | longevity |
| hug | loss |
| illness | love |
| kiss | loyalty |
| laptop | luxury |
| learning | marriage |
| meat | motivation |
| meeting | neglect |
| necklace | nutrition |
| nicotine | organization |
| noise | ownership |
| office | passion |
| painting | peace |
| party | poverty |
| perfume | presentation |

Table 2: Source Word Table (cont.)

| Concrete Words | Abstract Words |
|---|---|
| phone | prevention |
| pollution | procrastination |
| property | progress |
| protein | prosperity |
| purse | recovery |
| report | relapse |
| run | risk |
| salary | scarcity |
| savings | security |
| sculpture | sharing |
| sex | solvency |
| sitting | stability |
| sky | status |
| smile | strategy |
| sofa | strength |
| sun | stress |
| sunglasses | success |
| television | task |
| vaccine | time-management |
| vitamin | trust |
| walk | value |
| watch | vitality |
| water | weakness |
| yacht | wealth |

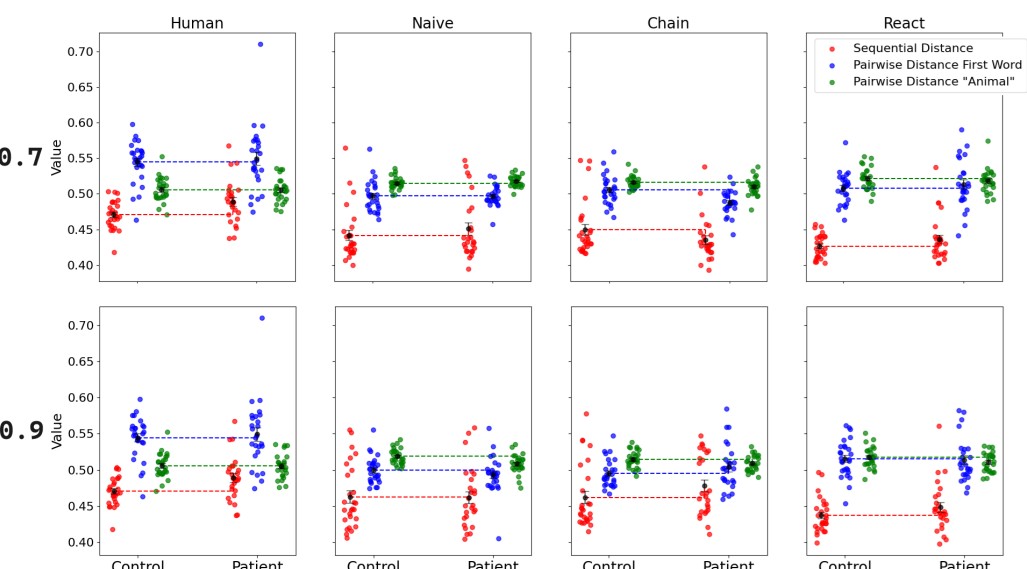

Figure 26: Same as figure 5 for temperature values of 0.7 and 0.9.

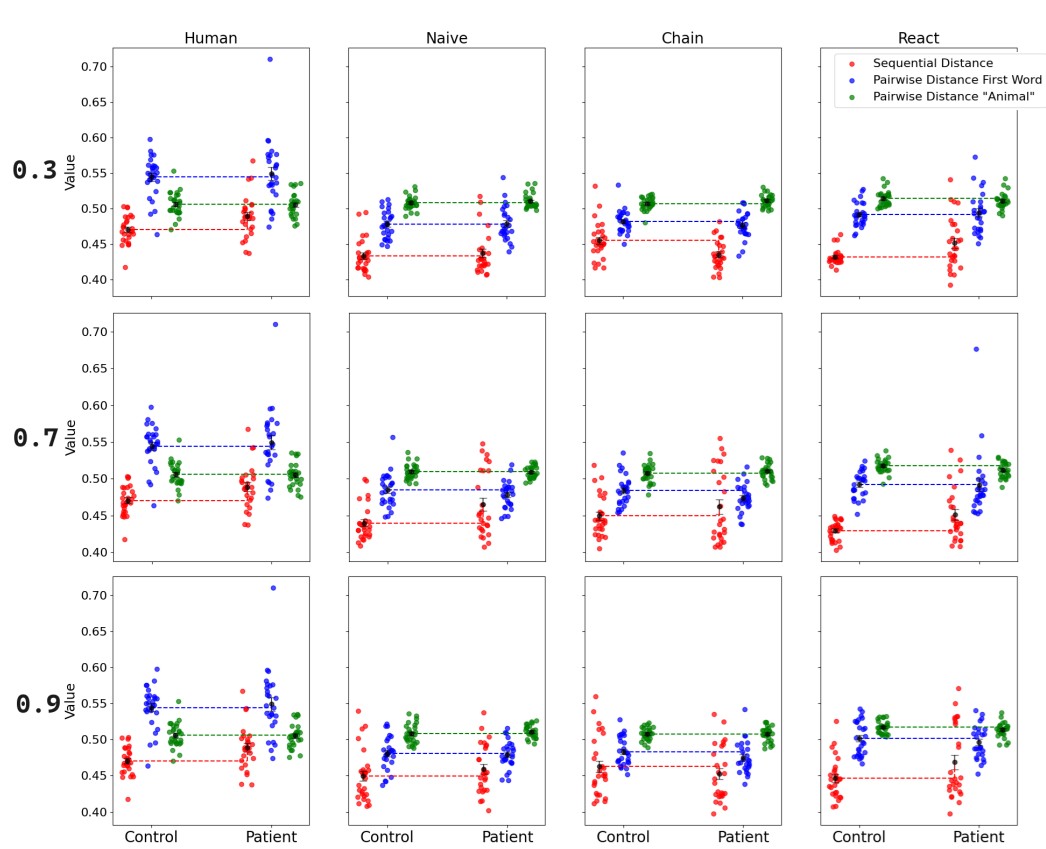

Figure 27: Same as figure 5 for Mistral, rows are temperature values and columns are prompting methods.

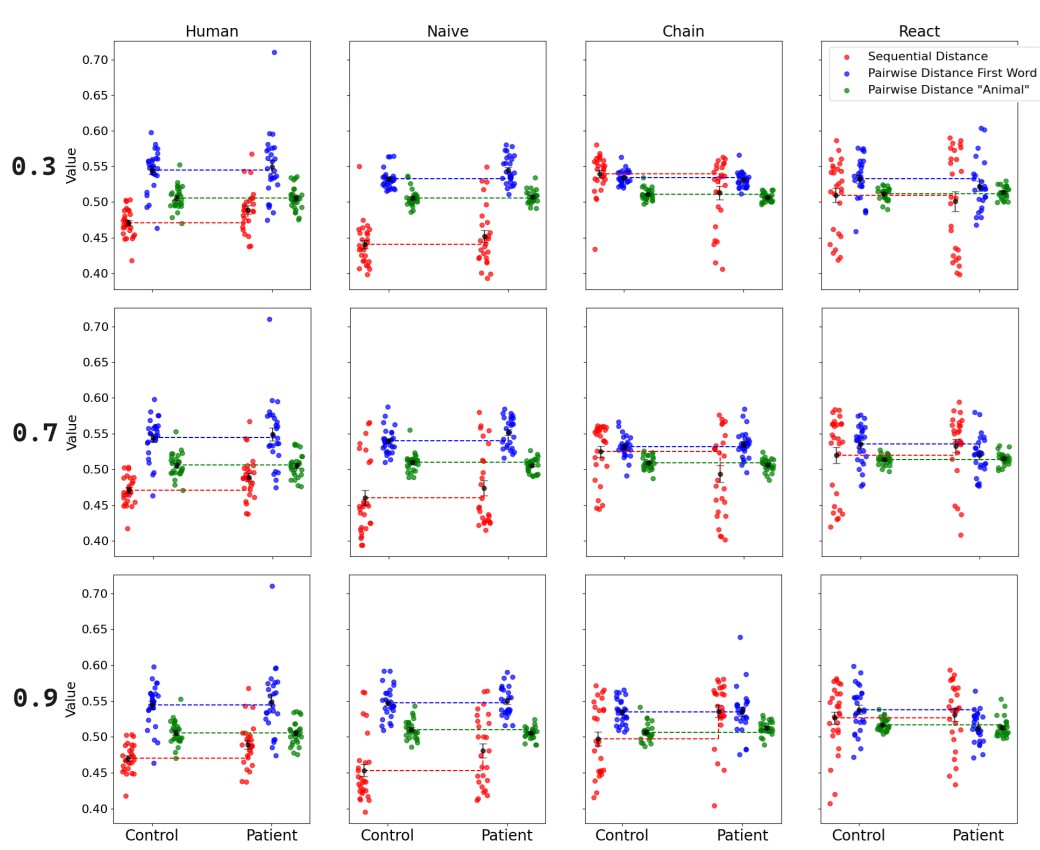

Figure 28: Same as figure 5 for GPT-3.5-Turbo, rows are temperature values and columns are prompting methods.

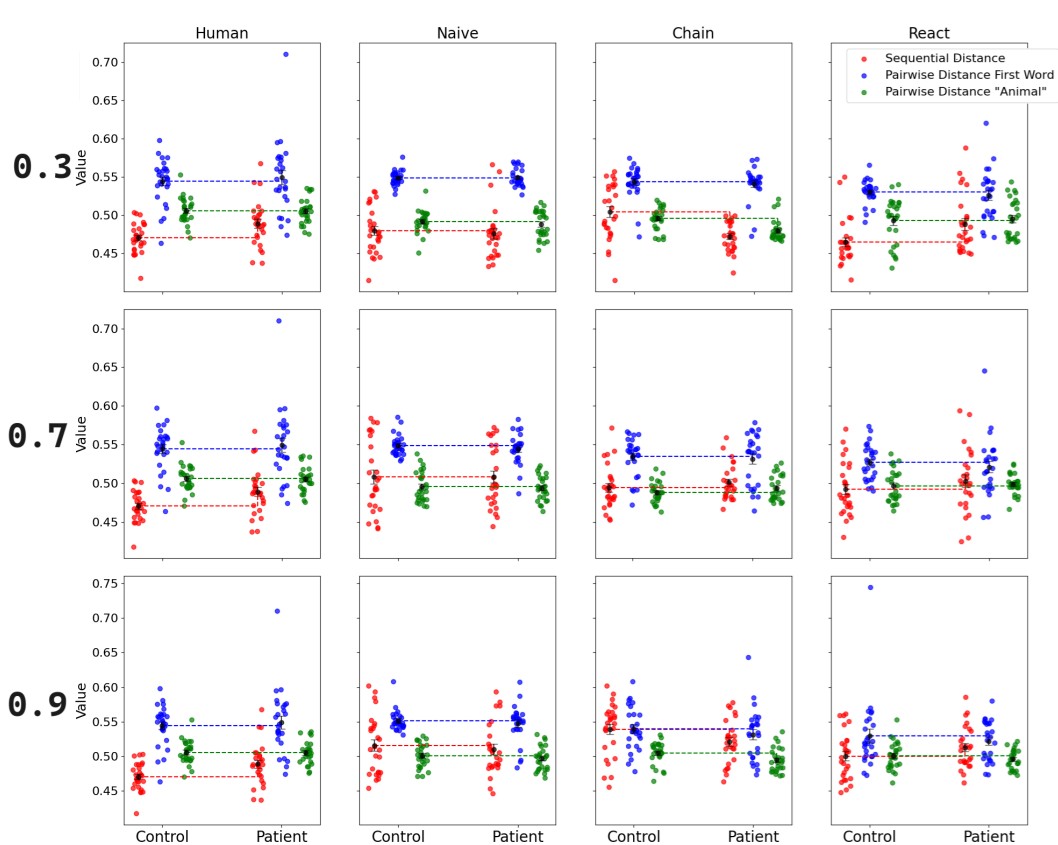

Figure 29: Same as figure 5 for Gemma, rows are temperature values and columns are prompting methods.

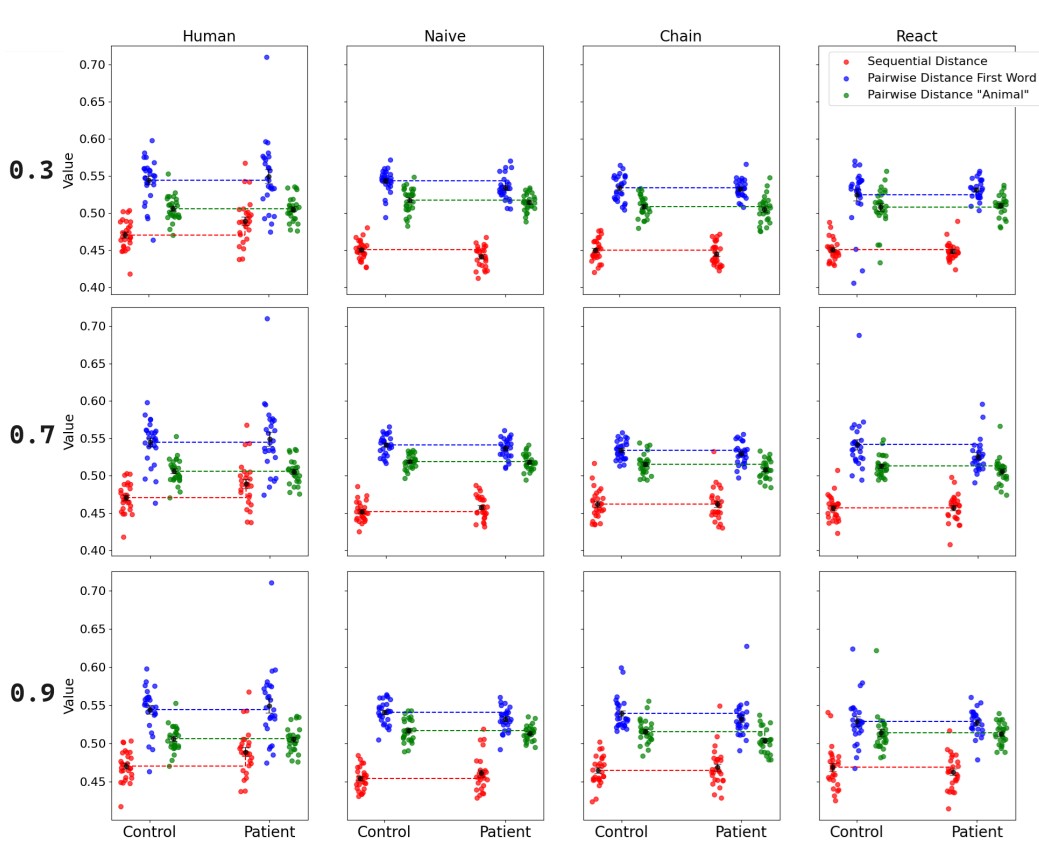

Figure 30: Same as figure 5 for Llama-2, rows are temperature values and columns are prompting methods.

## A.2 STATISTICAL ANALYSES OF CASE STUDY

Patients with schizophrenia exhibited a larger average semantic distance traversed through semantic space ($t(51) = 2.62$, $P = 0.01$, two sample $t$-test, two-tailed). No differences between patients and controls were observed for the first generated word or for the category animal (all $Ps > 0.5$). This pattern was not fully captured by Dolphin with React, since the difference between control and patient failed to reach significance in the averaged semantic trajectory traversed: $(51) = 1.24$, $P = 0.22$, two sample $t$-test, two-tailed. Furthermore, Dolphin with React elicited larger semantic distances when prompted as a patient in the first generated word, a pattern not observed in the human data ($t(51) = 2.14$, $P = 0.04$, two sample $t$ test, two-tailed), while the pattern for category animal was non-significant ($P > 0.5$), in line with the human data. When analyzing the other models, Gemma was the only one capturing the rank order between sequential distance, pairwise distance, and distance of "animal". Furthermore, React-prompted Gemma and Mistral with temperature 0.3 are able to reproduce the human data, with larger average semantic distances traversed for patients (both $Ps = 0.01$, two sample $t$ test, two-tailed), and no significant differences observed for first generated word and animal category measures (all $Ps > 0.2$). Future research should establish best practices for model-selection when simulating data from LLMs.

