# OpenReview forum: "Large language models as windows on the mental structure of psychopathology"
_ICLR.cc/2025/Conference — Submitted to ICLR 2025_

### Official Review · Reviewer_iDDR · 2024-10-31

**Soundness:** 1
**Presentation:** 2
**Contribution:** 2
**Rating:** 5
**Confidence:** 4

**Summary:**

The paper explores the potential of LLMs as tools for investigating the mental structure of psychopathology. The authors demonstrate that LLMs can be prompted to generate text responses that score highly on standard psychopathology questionnaires. They then analyse the semantic structure of the generated text using word embeddings and find that different psychopathologies are associated with distinct semantic structures. The authors argue that this approach could lead to the development of new hypotheses about the relationship between mental structure and psychopathology.

**Strengths:**

- LLMs are becoming a part of our society and it is interesting to see work which attempts to elicit their use for benefit of society.

- Some findings consistent with empirical observations- the results for depression and anxiety are consistent with previous empirical findings, demonstrating the potential of LLMs to simulate certain aspects of psychopathology.

**Weaknesses:**

- The abstract and introduction of the paper emphasize the importance of representations in understanding the inner workings of the mind. However, representations don’t seem to play any significant role in the experimental setup and analysis of the paper!  The prompted outputs seem to be used instead. A more direct approach comparing patient neural activation patterns with LLM embeddings may be needed to strengthen the claims.

- Line 25-26 talks about how lexical output patterns of LLMs are a proxy of their internal representations. However, this is very erroneous claim and much research (Turpin et al 2023 etc) has shown that models’ output as a function of their decoding strategy is not reflective of their internal representations.

- There seems to be a fundamental circularity in the experimental setup where we prompt models to generate psychopathic language and then we analyse the manipulated output to analyse the link between mental representation and psychopathology. But we specifically prompted models to behave this way and have these signatures that we look for later. It would be much more useful if we could use some kind of representational tuning of models with psychopathological representations and then analyse their output.

- There also need to be more models included to further strengthen claims (but this issue can only be addressed once the fundamental concerns highlighted before are fixed).

**Questions:**

- The paper's experimental setup needs to be improved in light of suggested weaknesses

**Details Of Ethics Concerns:**

The study is not sufficiently well structured and planned. At this stage, the paper's findings could be easily misinterpreted and misused by those without a deep understanding of both LLMs and psychopathology.Prematurely applying the study's findings to real-world diagnostic or treatment settings could lead to harm to individuals with mental health conditions.

---

> ### Author Response · Authors · 2024-11-22
>
> We thank the reviewer for her/his time dedicated in reviewing our paper, and provide point-by-point responses below.
>
> 1. "The abstract and introduction of the paper emphasize the importance of representations..."
>
> -> Only large models can generate the type of impersonation that we aim to study in this work. Models that prohibitively do not allow an analysis of internal representations (even in some case model that are not accessible due to proprietary reasons). Just as is the case in human neuroscience, researcher turn to indirect measures of the mental structure. One such measure is the "geometry" of lexical outputs, which as been shown, in many instances to nicely correlate with that of internal neural representations (e.g., Vives et al., 2023, Nature Human Behavior). Therefore, neuroscience has shown that lexical output patterns are a good proxy of internal representations. In our work, we do not care if LLM lexical output are a good indicator of their internal representations, this is not the aim of this work. More importantly, our work focuses in evaluating whether LLMs induced with a given psychopathology generate interesting lexical patterns that can provide human mental structure insights for clinicians in order to go delve deeper and validate potential interesting lexical patterns in humans. Hence the logical sequence is: 1) we can induce LLMs with psychopathology, 2) these induced LLMs generate lexical output patterns that resemble that of human with psychopathology, 3) human lexical output patterns are a good proxy for human mental structure, 4) hence, LLMs can help discover interesting lexical patterns in psychopathology that can then be tested by clinicians.
>
> Again, we must emphasize that we are not stating that lexical outputs in LLMs are a good proxy of their internal representations, but they can replicate human lexical outputs, and those have been shown to be a valid window on internal representations. The title of our paper does not state that LLM internal representations are a good proxy of human mental structure. We simply stated that LLMs are a good tool to evaluate the mental structure of psychopathology; which we claim to demonstrate in the paper.
>
> 2.  "Line 25-26 talks about how lexical output patterns of LLMs are a proxy of their internal representations."
>
> -> We agree with the reviewer that this is "an abuse of notation", and are perfectly willing to remove that sentence. In fact, we agree with the reviewer that LLM lexical outputs do no necessarily represent their internal representations. As discussed in the previous response, our claim is different. Note however, from being very acquainted with the Turpin et al. paper, that their work does not evaluate the relations between lexical outputs and internal representations. They mainly focus on showing that chain-of-thought responses can be biased in a way that generates a response that is not faithful to the true underlying reason for the response. Whether or not the internal representations are not aligned with the lexical outputs is another issue.
>
> 3. "There seems to be a fundamental circularity in the experimental setup..."
>
> -> We respectfully disagree with the reviewer. We prompt LLMs to impersonate psychopathology, and respond to a set of task. The lexical output of these tasks reveal signatures that can be found in those of humans that display the same psychopathology. In our opinion there is no circularity here. This is very similar than the plethora of impresonation work in the literature. The method proposed by the reviewer is interesting, but we lack the computational capacity to evaluate such a proposal. However, we will discuss this interesting proposal in the future work section of our paper.
>
> 4. "There also need to be more models included to further strengthen claims (but this issue can only be addressed once the fundamental concerns highlighted before are fixed)..."
>
> -> We can only test some models, and our choice tried to cover what we believe is a wide range of models. Our analysis is thourough and involves 23 supporting figures, exploring distinct prompting methods and temperatures. There is also a clear monetary limitation to the amount of simulation we can perform with other potential models (e.g., Claude 3.5 Sonnet).
>
> We again thank the reviewer for his/her comments, and hope we have addressed them.

---

> > ### Comment · Reviewer_iDDR · 2024-11-27
> > **Response to Authors**
> >
> > Human neuroscientific studies are different from using LLMs as windows into psychopathology and interpretability concerns remain. I would like to increase my score slightly in light of willingness to remove the emphasis on representations.

---

### Official Review · Reviewer_oJmx · 2024-11-02

**Soundness:** 3
**Presentation:** 4
**Contribution:** 2
**Rating:** 5
**Confidence:** 4

**Summary:**

The authors propose Large Language models (LLMs) as windows into the (human) mental structure of psychopathology. They prime models using instructions and descriptions of various mental disorders such as depression, schizophrenia, apathy, and others, before making them generate answers to the corresponding clinical questionnaires. They show that most tested models’ answers score high on the respective tests and above the diagnostic reference threshold. They then measure the impact of these primes on a word association task and show correlations with empirical results observed in humans. Finally, they finish with a case study comparing model outputs with the outputs of patients diagnosed with schizophrenia, again based on word comparisons.

**Strengths:**

The topic is very interesting. The experiments allow one to explore to what degree mental disorders can be tied to language and, therefore, be instilled and detected in LLMs as measured by the corresponding clinical questionnaires.

The article is well-written (apart from the grammar and spelling mistakes), well-structured and rich with empirical data and intuitive graphics. The overall presentation of results is very good.

The results show interesting correlations between the LLM output behaviour and what is expected to be observed in human patients diagnosed with the respective mental disorder. The general direction of research promises to reveal more interesting insights about how language is connected to the diagnosis and presence of mental disorders, which are separate points a priori.

The authors compare several models across 100 runs which broadens the experimental data and increase the representativeness of their results

The authors take care to delineate what their results do and do not show or implicate, which is especially important when dealing with sensitive topics like mental disorders and their diagnosis.

**Weaknesses:**

From an outside perspective, one could argue that the results show the possibility of priming models to mimic (and I am using this term in a strictly technical sense) the language behaviour observed in human patients with mental disorders. This can either display how strongly the field of psychopathology is bound to language or the limitation of questionnaires to fully capture it. To be fair, the authors never claim anything they did not substantiate with results, but my general concern is about how relevant this is regarding the understanding of psychopathology. Their approach certainly helps formulate new hypotheses to explore the language dynamics correlating with mental disorders and may help differentiate the latter from other human behaviours. However, the results only show post hoc correlations of answers but not evidence that these to-be-generated hypotheses can be meaningful.

More precisely, the authors analysed many representations of the LLM output (words that make up the answers) but not the output representation (the hidden states generated by the underlying transformer that shape the probability distribution from which tokens are sampled). The correlations displayed are certainly interesting as they seem to align the response behaviours of models and humans, but there is no justification for this correlation to be meaningful.

If, for example, during pre-training, an LLM encountered enough material about clinical tests and the answers correlating with the presence of a mental disorder, it may just display these correlations in its response behaviour. If the authors could instead use their approach to design a concrete hypothesis (which is the intended contribution of their article) and verify it using human data (which may not necessarily require the collection of new data), I would be more convinced that the displayed connections are meaningful. In essence, I think it is necessary to test how human response behaviour in a novel setup aligns with the LLM's response behaviour and not vice versa, as we cannot distinguish the degree to which models only imitate what they learned during training.

I am unsure whether the authors used the foundation models or the instruction-tuned versions for their experiments. This information needs to be added for clarity as instruction-tuned versions have been trained to adhere to (specially formatted) system prompts that can influence the impact of instructions on the generation behaviour.
Also, as the experiments require no fine-tuning, adding more modern iterations to the model pool (such as GPT-4/o/o-mini or Llama-3, which was published several months ago) seems promising. In this context, it would also be interesting to investigate how the scores scale with parameter size - has this been considered? Since most LLMs nowadays are based on the Transformer architecture and performance mainly depends on parameter size and data quantity/quality, scaling laws for any emerging capability are desirable.

The authors generate answers using three different temperature values, but other generation methods, such as beam search and contrastive search, are not explored.

Figure 2 shows the pathology scores for primed and unprimed models and the diagnostic reference threshold. Why are the "No Pathology" bars different across the three prompting techniques? For example, for Social Anxiety, we see decreasing scores for Mistral -> Dolphin -> Llama when using Naive prompts but increasing scores for the React prompts. From my understanding, the "No Pathology" results are not influenced by any prompt template, so why do we observe these contradicting trends when averaging the performances over 100 simulations? Likewise, the "No Pathology" scores crossing the diagnostic reference threshold seem to depend on the prompting technique.

In general, why do the "No Pathology" scores vary so much across the different prompting techniques when averaging them over 100 simulations and using a low temperature of 0.3? Can the authors explain this?

The heatmaps in Figs. 8-12 show that there exist significant "side-effects" when inducing a specific mental disorder pattern using any of the prompt templates. This has been acknowledged in the text and linked to data observed in humans (testing positive w.r.t. more than one mental disorder questionnaire), but I feel that it also limits the scope of the approach. The red fields make it easy to spot whenever a perfect score of 1.0 is achieved, but they also slightly divert the attention from the other fields that show a value of 1.0. For example, in Fig. 8, several off-diagonal tiles also show a value of 1.0, and many values change drastically across the different prompting templates. Which of these values are meaningful? Does it make sense to evaluate the minimum/maximum?

Also, more information is required: the authors cite some works in the paragraph starting in line 280, but it is unclear which of the observed patterns in the normalised Lickert scores are mirrored by human behaviour. A more detailed analysis is needed to convince readers that these patterns are meaningful (it would likewise be appropriate to move at least one of the figures to the article's main body).

The cosine similarity between two vectors is a common metric to compare how aligned they are (helping to compare vector representations of semantic concepts). However, the simplex volume needs more (mathematical) motivation and an explanation for why this expression is well-defined.

The results displayed in Figure 5 seem inconclusive. As can be seen by the (red) outliers in both human and LLM "Patient" data, determining the mean seems less adequate than, for example, the median (also, the standard error bars are difficult to detect, even when zooming in - increasing the thickness of the bars or the transparency of the dots may improve the plots). Instead of this quantitative analysis, I think it would be more attractive to qualitatively explore what may have caused the outliers to be significantly higher in human and LLM data. This could lead to formulating a concrete hypothesis (based on LLM data) for further testing and verification (with human data).

Overall, the article is good and a solid basis for future work, but I think the experimental results, albeit interesting, are too superficial. More evidence is needed to show why this approach is meaningful. However, I am open to being convinced otherwise and to change my rating.




Minor:

What is “PT-prompting” in Figure 1? Does it refer to “psychological traits”-prompting?

The quality of the figures needs to be improved, especially for Fig.s 8-12.

Grammar and spelling mistakes need to be corrected, for example:

Line 020/021: “could be viable tool to” -> “could be viable tools to”

Line 051/052: “one of the basis of” -> “one of the bases of”

Line 313/314: “To evaluate representational” -> “To evaluate the representational”

Line 394/395: “sources words” -> “source words”

Line 396/397: “generate words embeddings” -> “generate word embeddings”

Line 430: “extract using” -> “extracted using”

Line 482/483: “novel hypothesis” -> “novel hypotheses”

Line 502: “green red” -> “red”

Line 510/511: “future as a ease-to-use of psychological tasks” -> best to rephrase that entire sentence

Line 518/519: “(Nour et al., 2023), measures.” -> “(Nour et al., 2023).”


Denoting the cosine similarity measure with “cos” is confusing.

Kappa and Delta need to be switched in lines 369 and 370.

In Figure 3, the RSA score heatmaps are at the top/middle/bottom (not left/middle/right). Also, the word “between” in the parentheses part needs to be removed.

**Questions:**

See Weaknesses.

---

> ### Author Response · Authors · 2024-11-22
>
> We thank the reviewer for the positive appraisal of our work. We also are grateful for the thorough and high quality revision of our work that will undoubtedly increase its quality.
>
> Before we respond to the comments, we thank the reviewer for spotting typos and small mistakes that will be revised in our CR-version.
>
> 1. "However, the results only show post hoc correlations of answers but not evidence that these to-be-generated hypotheses can be meaningful."
>
> -> We fully agree with the reviewer for the first part of our paper. However, we do show the value of our work in a case study that involves schizophreny above and beyond simple correlations. Our hypothesis is that LLMs induced with schizophrenic behavior should display similar lexical structure patterns than those of human patients, which is exactly what we observe in section 5 of our manuscript. Thereby, we do believe that our results do in fact show "evidence that these to-be-generated hypotheses can be meaningful", and thus that our method is meaningful at least in inferring potential qualitative patterns.
>
> 2. "recisely, the authors analysed many representations of the LLM output (words that make up the answers) but not the output representation..."
>
> -> The reviewer is right again here. Nonetheless, it is worth mentioning that complex impersonation, as the ones studied in our work, typically emerge in LLMs whose internal representations are extremely hard to analyze (and in many cases not even accessible). We believe the correlations (although admidetly exploratory) do provide interesting insights worth reporting.
>
> 3. "If, for example, during pre-training, an LLM encountered enough material about clinical tests..."
>
> -> As replied in the previous comment, we do believe that Section 5 tackles exactly what the reviewer raises. Here, schizophrenic patients generate a list of animals and, compared with control humans, this list displays strong semantic expansion. This is a concrete hypothesis that we tested with our method, and was confirmed with LLMs induced with schizophreny. Note that the original paper did not report the list of animals, and therefore LLMs were not exposed to these type of data. The data is found in an excel file and cannot be used for monetary purposes. Therefore, we believe to convincingly show that our results do not emerge from pre-training imitation.
>
> 4. "I am unsure whether the authors used the foundation models or the instruction-tuned versions for their experiments..."
>
> -> All but Dolphin are instruction-tuned versions; we will add this information in the CR-version of the paper. Moreover, the parameter size is unlikely to play a major role here. Indeed, the most promising model (i.e., Dolphin), was substantially smaller than other models. Future work should consider other dimensions, such as alignement, in the ability to capture human data.
>
> 5.  "The authors generate answers using three different temperature values..."
>
> -> We thank the reviewer for this suggestion and will explore this initiative in future work given the already very loaded content of our current manuscript.
>
> 6. "Figure 2 shows the pathology scores for primed and unprimed models and the diagnostic reference threshold..."
>
> -> The prompts are not extactly the same for the no pathology across the three methods. This can potentially explain the differences in the no pathology values. To maintain internal consistenty within each prompting method we generated the no pathology prompt following the same lexical input pattern than the one used for pathologies. This systematic differences in input pattern between prompting methods may generate the variablity observed, even with small temperature values.
>
> 7. "The heatmaps in Figs. 8-12 show that there exist significant "side-effects" when inducing a specific mental disorder pattern using any of the prompt templates..."
>
> -> We treat these analyses are exploratory, and could provide interesting insights for clinicians. The aim of those heatmaps is to show that, in most cases, inducing a specific psychopathology does in fact influence the targeted psychopathology, although showing also interesting "side effects".
>
> 8. "Also, more information is required..."
>
> -> We agree with the reviewer but are unfortunately bounded by space limitations. Future work will focus on delineating in more detail which lexical feature in humans are best capture by LLMs.
>
> 9. "However, the simplex volume needs more..."
>
> -> Cosine similarity can sometimes inflate the semantic expansion value when many vectors point in one direction and another points towards another direction, and potential effect which is better controled by the simplex volume measure.
>
> 10. "The results displayed in Figure 5 seem..."
>
> -> We believe that ReAct prompting captures all but one qualittive aspect of data. Indeed, all the relative differences with the control groups are reveavled by our LLM simualtion. For consistency, we did the same analyses as the original paper.

---

> > ### Comment · Reviewer_oJmx · 2024-11-25
> >
> > **Thank you for responding to my review and the mentioned criticism. My points of critique have not been addressed adequately, nor were my questions answered. I will formulate a new collection of points and questions for the authors to address below and would greatly appreciate a new response.**
> >
> > *-> We fully agree with the reviewer for the first part of our paper. However, we do show the value of our work in a case study that involves schizophreny above and beyond simple correlations. Our hypothesis is that LLMs induced with schizophrenic behavior should display similar lexical structure patterns than those of human patients, which is exactly what we observe in section 5 of our manuscript. Thereby, we do believe that our results do in fact show "evidence that these to-be-generated hypotheses can be meaningful", and thus that our method is meaningful at least in inferring potential qualitative patterns.*
> >
> > You reference the schizophrenia case study as the result that shows your work’s value beyond correlations while writing in line 460/461: “Note however that this difference did not reach statistical significance, contrary to what is observed in human data [...]”. Moreover, you write in line 431 that “Dolphin was able to capture this qualitative pattern”, but then in line 466/467: “Dolphin could not capture the rank of all the distance values.” in regards to other experiments (first generated word and animal word). This seems like cherry-picking results, which does not convince me. Yes, Gemma was able to mirror the rank order, but its ReAct responses also achieved a near-threshold score in Fig. 2 for schizophrenia, and the other results displayed in Fig. 29 do not convince me either.
> > (By the way, the word “schizophreny” does not exist. You use the correct word, “schizophrenia”, in the article but never in your response.)
> >
> > *-> As replied in the previous comment, we do believe that Section 5 tackles exactly what the reviewer raises. Here, schizophrenic patients generate a list of animals and, compared with control humans, this list displays strong semantic expansion. This is a concrete hypothesis that we tested with our method, and was confirmed with LLMs induced with schizophreny. Note that the original paper did not report the list of animals, and therefore LLMs were not exposed to these type of data. The data is found in an excel file and cannot be used for monetary purposes. Therefore, we believe to convincingly show that our results do not emerge from pre-training imitation.*
> >
> > Just because the exact data are not available on the internet does not mean that similar data are present, which could nevertheless influence model responses. R1 uses the word “contamination”. This does not always work directly but can result from proxies.
> >
> >
> >
> >
> > *-> All but Dolphin are instruction-tuned versions; we will add this information in the CR-version of the paper. Moreover, the parameter size is unlikely to play a major role here. Indeed, the most promising model (i.e., Dolphin), was substantially smaller than other models. Future work should consider other dimensions, such as alignement, in the ability to capture human data.*
> >
> > This strikes me as odd. The only model that was not instruction-tuned is the most promising model in combination with the three prompting techniques. In particular, Dolphin was never trained with special tokens that signal instructions and system prompts. How can the only model that was not trained to assume a persona be the most promising at imitating a persona that shows specific psychopathological traits? Do you have any explanation for this?
> >
> > *-> The prompts are not extactly the same for the no pathology across the three methods. This can potentially explain the differences in the no pathology values. To maintain internal consistenty within each prompting method we generated the no pathology prompt following the same lexical input pattern than the one used for pathologies. This systematic differences in input pattern between prompting methods may generate the variablity observed, even with small temperature values.*
> >
> > What are the differences between the three methods for the no-pathology prompts? Did you explain these in the article? If we compare the OCD results across the three strategies, we notice that the no pathology prompts show barely any effect for the last case (which is what we would expect if there is no psychopathology induction at work) compared to the first and second case, where we do notice some models crossing the threshold. Does it make sense to consider the performance score deltas in Fig. 2 instead to detect differences?

---

> > > ### Comment · Reviewer_oJmx · 2024-11-25
> > > **-continued-**
> > >
> > > *-> We agree with the reviewer but are unfortunately bounded by space limitations. Future work will focus on delineating in more detail which lexical feature in humans are best capture by LLMs.*
> > >
> > > So you agree that “[...] it is unclear which of the observed patterns in the normalised Lickert scores are mirrored by human behaviour. A more detailed analysis is needed to convince readers that these patterns are meaningful”, but you do not add this analysis to your work, be it to the main body or the Appendix (to which no space limitations apply)?
> > >
> > >
> > > *-> Cosine similarity can sometimes inflate the semantic expansion value when many vectors point in one direction and another points towards another direction, and potential effect which is better controled by the simplex volume measure.*
> > >
> > > The cosine similarity is a measure that maps two vectors of the same dimension to a real number. “Many vectors” are only present in your average in equation 1. So you criticise using the average of cosine similarities, which you introduce as a metric in the first place.
> > >
> > > Moreover, you explained a weakness of equation 1, which a priori is not an argument for using the simplex volume. This is precisely my criticism that I repeat here: “However, the simplex volume needs more (mathematical) motivation and an explanation for why this expression is well-defined.” R1 also criticises this point in his original review and his follow-up response.
> > >
> > >
> > > *-> We believe that ReAct prompting captures all but one qualittive aspect of data. Indeed, all the relative differences with the control groups are reveavled by our LLM simualtion. For consistency, we did the same analyses as the original paper.*
> > >
> > > Which qualitative aspect of the data does ReAct prompting not capture?

---

### Official Review · Reviewer_hqwG · 2024-11-03

**Soundness:** 3
**Presentation:** 3
**Contribution:** 3
**Rating:** 6
**Confidence:** 3

**Summary:**

The authors proposed a way to leverage LLMs to generate dialogues, which can be used to evaluate whether it shows psychopath or other symptoms. The authors also foresee that this method can be used in practical fields like Cognitive Behavioral Therapy. The authors conducted thorough analysis with different open models, and conducted statistical significance measurement.

**Strengths:**

- The authors performed thorough design and analysis on the performance of the framework with different open models
- The authors conducted analysis on statistical significance

**Weaknesses:**

- Lack of literature reviews on similar approaches in psychology fields in top conferences. The cited paper "The empirical structure of psychopathology is represented in large language models" did not show up in top conferences
- The authors only shows that there is ways for LLMs to participate in the psychology questionnaires and get scores. More examples on potential applications (e.g. in CBT) can be discussed.

**Questions:**

- Can you do more literature reviews on top conferences for similar approaches in psychology applications?

---

### Official Review · Reviewer_D1xL · 2024-11-04

**Soundness:** 1
**Presentation:** 2
**Contribution:** 1
**Rating:** 1
**Confidence:** 4

**Summary:**

The paper proposes to examine LLM representation as a parallel with certain psychopathologies, such as apathy, depression, or anxiety.
LLMs are first prompted in various manners to exhibit such conditions, and then an assessment of their internal structure is done via prompting for word associations and measuring distances with Glove embeddings. Then these metrics are compared to human subject data collected in previous study to see how much of it is explained by the LLM performance.

**Strengths:**

- I think that it's timely project to examine the correlations between human and LLM behavior, and the idea to correlate that carefully against existing data is interesting.

**Weaknesses:**

Despite the timeliness of the work, I have major concerns about it in its current state.

* **Methodology** - Most importantly, I doubt the methodology, and hence doubt also the results obtained from it.

First, there seem to be some implicit parallels drawn in this work between LLMs and human subjects, but this is not made explicit. Instead the paper refers to the "structure" of LLMs, which is unclear to me. There's a point to be discussed that the inner representation of the LLMs are its weights, in which case, it's the same LLM when different prompts are made, which I think contradicts the paper approach that the LLM personifies different subjects with a different prompt. In contrast, it's possible to claim that the representation of the LLM is composed of its weights + the activations induced from the prompt, which would better fit the paper claim, but none of this non-trivial point is discussed or argued by the paper.

Second, I find the initial experiments (Figure 2, and the evaluation of co-morbidities) highly questionable, and hence as the paper states this is "an important result that forms the basis for the following analyses", I consequently doubt the following results. I doubt figure 2 because of other possible explanation beyond an actual "underlying psychopathology" in LLMs (Line 285), e.g., since all of these exams are available online, it's very probable that they appeared in the LLM pretraining corpus (contamination), which actually I think aligns with the paper's finding that LLama, Mistral show high scores "even when prompted with no pathology" (Line 260). The correlation between different conditions is an interesting way to alleviate these concerns, but here too I disagree with the chosen methodology, the paper compares the different models relatively to one another and chooses the one which better explains human data: "Dolphin displays
the highest correlation (0.59..." (Line 301). I don't think that these measures should be taken in a relative manner. Instead, I think that models should clear some absolute bar to be considered as correlated with humans. For example, how do human subject correlate on this scale? Is 0.59 significant? Also, I don't know if RSA is a common way to compute such similarities, the paper doesn't cite a source for this comparison.

Finally, I disagree with the paper's claim that the experiments conducted in Section 4.1 are relevant proxies for "representational structure". Equations (1) and (2) aren't motivated in literature or in concrete evidence that they reflect inner representation (a terms which wasn't explicitly defined, as I mentioned above). Even if justified, I find the use of Glove embeddings highly questionable -- from what I understand it is taken as a sort of "gold" representation for semantic diversity, which isn't justified, explained, or evaluated. Instead, I would work on justifying why equations (1), (2) are correlated with some explicit inner representation, and then evaluate it manually for at least a subset to understand how well glove approximates this score.

* **What are the actual takeaways?** -
I struggle with understanding what would be the takeaway of the paper? How would it benefit future research? The paper says in the conclusion that "We suggest that our method can help generating novel hypothesis regarding the between link mental structure and psychopathology, in a cost effective and scalable way" (Line 482). How? Is it possible to flesh this out more?

* **Appropriateness for ICLR** - Related to the previous question, I wonder who would be the appropriate audience for this paper. If the conclusions benefit psychologists, then perhaps it's a better fit at that kind of venue? Relatedly, there are many claims made in the paper about psychology (e.g., in Section 3.1), which I'm not qualified to review, since I don't have any expertise in this area.

**Questions:**

- What's the license of the data used in this work? Did the subjects give their consent for this kind of use?

---

> ### Author Response · Authors · 2024-11-22
>
> We thank the reviewer for her/his time invested in reading our work and the suggestions proposed.
> However, we are surprised by the weaknesses raised by the reviewer and in particular the score given to our work which basically suggests there is no value to our work. We reply to the comments below:
>
> 1. Methodology:
>
> 1A. Our work is in line with the many studies looking at how LLMs can be used in impersonation scenarios (as we point out in the paper). A very similar paper to ours, methodologically speaking, but in another domain, i.e., that of personality traits, performs a similar impersonation method. To check that the impersonation is succesful they also evaluate these with standard scales. That paper was awarded as a spotlight paper in NeurIPS 2023 (Jiang et al.). This initial induction phase is just a sanity check on the ability to push the model towards a "space" that generates lexical outputs that would resemble that of a person diagnosed with a certain psychopathology. Whether the dataset contains the responses to scales is irrelevant to our research question. Here, we first check that we can simulate a psychopathology-like lexical output, and then we observer whether these lexical outputs can be linked to those generated by patients and only in terms of distance metrics in lexical outputs. This second step is the crucial novelty of our work.
>
>
> 1B. If the reviewer believes that the correlation with human data should pass an absolute bar, I invite him/her to provide a number? We belive that 0.59 is an extremely high correlation value given the task at hand. Moreover, there exists no absolute value in the literature and this analysis is the first of its kind, highlighting the novelty of our work.
>
> 1C. With respect to the structural representation of LLMs, very similar to the many studies in cognitive science (which we cite), we approach this question via an indirect measure, i.e., distance of lexical outputs. This is a common approach in the field and we have cited several papers that look at the structure of internal representations in cognitive neuroscience from that perspective. In cognitive neuroscience, the RSA analysis is one of the most common methods to evaluate whether lexical (or neural for that matter) representations are similar (see Vives et al. 2023, Nature Human Behavior). In the same vein, the distance metrics we provide are common methods as well (again see Vives et al. 2023, Nature Human Behavior).
>
> 1D. The use of GloVe representations allows to generate numerical analysis on the distance between lexical outputs. Internal representations of LLMs are seldom accesible, especially with LLMs capable of generating complex impersonations.
>
> 2. Takeaways:
> As we explain in the text, potential psychiatric treatments can emerge from understanding the structure of mental representations in psychopathology, and how these structures may differ from those of people not diagnosed with specific psychopathological conditions. However, evaluating these structures comes at great cost, both in time and money. Here, we show that LLMs are a promising tool to capture the lexical structures present in psychopathology in a easy, cost and time effective way. Our second paragraph in the introduction delineates why our work matters, and its consequences for therapy.
>
> 3. We would not have sent our paper to ICLR if we did not think it fits with the purpose of the conference. We sent the paper to the applications of LLMs in neuroscience. We belive our work fits well that theme.
>
> Reply to question:
>
> The human data we used in this paper are open-source data and can be used in any future work. We cite the original papers and provide the links to the data.
>
> We hope to have addressed the concerns of the reviewer.

---

> > ### Comment · Reviewer_D1xL · 2024-11-24
> >
> > I'd like to start by saying that my review did not mean to suggest that there's no value in the work, only that I find major flaws in it. I'm afraid that the response didn't answer my concerns, I'll try to rephrase them below and respond to the author response. I hope that these concerns are addressed in future versions of the paper.
> >
> > 1A. I don't finding pointing to another paper an answer to my questions. Specifically, I'm concerned about contamination, which wasn't discussed in the paper or in the response. I find some disparity between this sentence from the paper which I cited in my review: "an important result that forms the basis for the following analyses" with regards to section 3.2 (INDUCING AND EVALUATING PSYCHOPATHOLOGY IN LLMS), and the response saying that "This initial induction phase is just a sanity check". I think that contamination can have great effect where LLMs would behave one way on known datasets, but differently on another set, hence harming the generalizability of the results.
> >
> > 1B. I do believe that systems should pass an absolute bar with human results, otherwise, take any random baselines, and choose the best performing system out of them, would that make sense? Besides, I think that the response also acknowledges that there should be an absolute bar -- "0.59 is an extremely high correlation value" -- as opposed to the paper, which didn't try to claim anything absolute about the results. I think that asking the review process to produce justification for this bar is shifting the burden of proof. If the paper proposes that there's equivalence between mental states in human and LLMs, it should convince the reader of this claim. One suggestion is to measure how a **human** participant's scores correlate against a group of their peers? Will that be similar to 0.59? If so, I would find this interesting.
> >
> > 1C. I think that the paper would benefit from a definition of "structure" in LLMs. Does that mean the weights? or the activations? Something else? This would help clarify much of the paper IMO.
> >
> > 1D. But Glove will inherently introduce errors, how do they affect the validity of the analysis? I think that a human evaluation would help answer this.
> >
> > 2. Takeaways: But if the intended audience is psychologists who should adopt LLMs, then I think that reaching out to psychology venues may also be beneficial.
> >
> > 3. Regarding the data - what is the license of the data? There are different levels of open source, and I think that specifying the exact license would be useful.

---

> > > ### Author Response · Authors · 2024-11-24
> > >
> > > Thank you for engaging in the discussion.
> > >
> > > 1A. Does that entail that all the impersonation papers are flawed? Can the reviewer explain how our paper differs from the plethora of impersonation papers in the literature?  As for all the impersonation papers, we prompt the models to behave in a certain way. Acsking an LLM to act politely can only come from having been pre-trained with polite examples. Critically and to completely clarify the concern of the reviewer, human (with or without psychopathology) responses to DSM questionaires are completely private and cannot be used in the pre-training of LLMs. The questionnaires themselves may be, but the results of human using them are fully private. Therefore, unless the LLMs we have used infriged direct privacy laws, there is no chance that these models have been pre-trained on data that contains human responses to these questionnaires. More importantly, even in the case that LLMs may have used some human data (which would be infringing privacy laws as just mentioned), the fact that we used scales that may be in the training set is irrelevant to the point of this paper. We aim to evaluate whether induced models can generate outputs that resemble that of humans diagnosed with certain psychopathology; which we do demonstrate in the paper.
> > >
> > > 1B. Again I invite the reviewer to tell us what that absolute bar should be. Random variables correlate to 0, is anything above or below that a good number? We do not think engaging the review process to set a value for that bar is a burden, we bleive this how papers get better. The reviewer proposes to:
> > >
> > > "One suggestion is to measure how a human participant's scores correlate against a group of their peers? Will that be similar to 0.59? If so, I would find this interesting."
> > >
> > > Does the reviewer suggest that such a measure should be the absolute bar? Wouldn't that bar be extremely high? In our work we are not concerned in showing that induced LLMs act just like humans, or can capture all types of correlations displayed by humans. We develop a thorough exploratory analysis, and then focus on demonstrating that LLMs can be an impressive tool to guide research. In fact, our paper shows that LLMs induced with schizophrenia can recapitulate lexical outputs from schizophrenics on a task that we know did not contaminate the pre-training; we believe this is an amazing result worth inferring that LLMs can be a great tool to explore more of these lexical structures.
> > >
> > > 1C. Thank you for this suggestion. Indeed, this may be confusing for people that are not from the field of (cognitive) neuroscience. There are many studies demonstrating that lexical output patterns are reflected in the internal neural representations in humans (see Vives et al., 2023). In our paper, we are not trying to get at the issue of the internal representations (i.e., in activation space, which would be the equivalent of neural space in humans) of LLMs; or even how the LLM lexical outputs related to those internal representations. We focus on whether impersonated LLMs can generate lexical outputs that are present in human psychopathology. We demonstrate this is the case, thereby proving the value of LLMs as tools that can guide clinical research and provide educated guesses on the potential lexical output patterns in psychopathology , and therefore on the mental structure (i.e., neural representations) of psychopathology (see Vives et al., 2023).
> > >
> > > 1D. We do not understand the suggestion of the sugegstion of the reviewer, what would the human be evaluating? Embedding the words using GloVe is merely a tool to generate quantitative results, using a human to evaluate the "proximity" of words is filled with biases, and does not allow to compute volumetric or distance measures. In any case, if GloVe introduces a bias, it is the same bias for all simulations. Again, I believe the reviewer is overweighing the importance of demonstrating that LLMs are similar to humans, they are not. We aim at showing that they can be a good tool for research in clinical psychology/psychiatry, and we prove that they are.
> > >
> > > Takeaways: the theme is applications of LLMs in neuroscience. Does the reviewer believe that our paper does not fit well that theme?
> > >
> > > Data: Thank you for this comment. Indeed, the data are perfectly usable for research, we will make this clear in the revision.
> > >
> > > We thank again the reviewer for engaging in the discussion.

---

> > > > ### Comment · Reviewer_D1xL · 2024-11-25
> > > >
> > > > 1A. The explanation here engages with the question of contamination and tries to explain why that may not happen, I think this should be added to the paper, and ideally also some perturbation experiments which could validate that this is the case.
> > > >
> > > > 1B. I suggested a bar - comparison to human behavior, as this is what we're trying to approximate. Even if that is extremely high and models don't reach this bar, then this still helps contextualize what this correlation means, and we know how close they are to achieving this, and will be meaningful for future work.
> > > >
> > > > 1C. I think this should be clarified in the paper.
> > > >
> > > > 1D. How would human psychologists evaluate the lexical patterns of human subjects? It's problematic to say that there's no way to evaluate the output of an automated systems like Glove, what if Glove does a terrible job on the tested words? Glove of course also introduces biases into the evaluation. To me this is one of the major flaws in this paper. Again, the burden of proving that an automatic system performs well is on the developers, not the reviewer. I don't understand how critically assessing the claims of the papers is "overweighing" them. This kind of response came up also with regards to my other comments (e.g., contamination, which after a discussion I think that helpful explanations were produced). Another point which I find missing from my original review is an explanation of Equations (1), (2).
> > > >
> > > > 2.  I would still appreciate to learn what is the license of the data. Can you explicitly state what is the license?

---

### Meta-Review · Area_Chair_EzZm · 2024-12-19

**Metareview:**

**Summary**

This paper push on the metaphor of considering LLMs as good representative of human brains. Hence, these models are used as subjects to be tested with psychopathology questionnaires.

**Strengths**

- The metaphor "artificial brain"-"wet brain" is extremely intruguing

**Weaknesses**

- The metaphor "artificial brain"-"wet brain" is extremely misleading as LLMs seem to be only Language Models and not more.


**Final remarks**

This is the kind of paper that may attract a lot of attention as it may push the boundary of the parallelism between machine and human minds/brains. Cognitive psychology uses machines to describe human minds. In this case, the term computational neural networks has generated the confusion. Indeed, LLMs seem to be good memorizers that have a hard time in generalizing. It is hard to compare these computational models with wet brains. If the LM is seen in its probabilistic form, it is hard to imagine that LMs may somehow represent the behavior of humans.

**Additional Comments On Reviewer Discussion:**

The authors failed to convince reviewers and also the AC that the metaphor is a good metaphor. In order to reach the point where LLMs can be considered more than Language Models, there should be a better understanding of how transformers use tons of data ingested.

---

### Decision · Program_Chairs · 2025-01-22

Reject